# LATENT PLANNING EMERGES WITH SCALE

**Michael Hanna**[*]
ILLC, University of Amsterdam
m.w.hanna@uva.nl

**Emmanuel Ameisen**
Anthropic
emmmanuel@anthropic.com

## ABSTRACT

LLMs can perform seemingly planning-intensive tasks, like writing coherent stories or functioning code, without explicitly verbalizing a plan; however, the extent to which they implicitly plan is unknown. In this paper, we define *latent planning* as occurring when LLMs possess internal planning representations that (1) cause the generation of a specific future token or concept, and (2) shape preceding context to license said future token or concept. We study the Qwen-3 family (0.6B-14B) on simple planning tasks, finding that latent planning ability increases with scale. Models that plan possess features that represent a planned-for word like *accountant*, and cause them to output *an* rather than *a*; moreover, even the less-successful Qwen-3 4B-8B have nascent planning mechanisms. On the more complex task of completing rhyming couplets, we find that models often identify a rhyme ahead of time, but even large models seldom plan far ahead. However, we can elicit some planning that increases with scale when steering models towards planned words in prose. In sum, we offer a framework for measuring planning and mechanistic evidence of how models' planning abilities grow with scale.

## 1 INTRODUCTION

LLMs succeed at some tasks that seem to require planning—reasoning about the steps needed to achieve a goal state—without explicitly verbalizing a plan. Understanding the extent of models' unverbalized planning is important: such *latent planning* could present AI safety risks, allowing models to engage in scheming without alerting external monitors (Balesni et al., 2024; Korbak et al., 2025). Despite this, empirical evidence regarding LLMs' latent planning remains limited.[1] Past work on latent planning is largely observational: studies show that future tokens or text attributes can be extracted from model activations (Pal et al., 2023; Pochinkov, 2025; Dong et al., 2025). Only recently has causal evidence for planning emerged, in closed models (Lindsey et al., 2025).

We argue that claims of latent planning must be based on causal, not observational evidence, lest we apply the "planning" label too broadly. We consider an LLM to engage in latent planning only if it possesses an internal representation of the planned-for token or concept $t$ that causes it to generate $t$; we call this *forward planning*. However, this representation must also cause the model to engage in *backward planning*, reasoning back from its goal $t$ to generate a context that accommodates it.

To understand how latent planning emerges with scale, we test 5 Qwen-3 models of increasing size on simple tasks that could involve latent planning, like completing "Someone who handles financial records is → a/**an** (accountant)"; we find that only models with 14B+ parameters consistently succeed. We then use feature circuits (Marks et al., 2025; Ameisen et al., 2025) to find the mechanisms that underlie models' abilities. We find that there exist planning features that represent future outputs like *accountant* and upweight relevant outputs like "an" (Figure 1). Moreover, although smaller models fail, they possess planning-relevant features that promote the correct answer.

We next have models complete rhyming couplets, where Lindsey et al. observed longer-range planning in Claude Haiku. We find that models employ a circuit that tracks information related to poetry, such as when a line is about to end, or what to rhyme with; however, even large models do not engage in backward planning. We then test intermediate planning abilities by steering models towards planned words in prose, and observe forward and backward planning, increasing

---

[*]Completed as part of the Anthropic Fellows Program.

[1]Explicit, verbalized planning, as in LLMs' chains of thought, is better studied (Kambhampati et al., 2024).

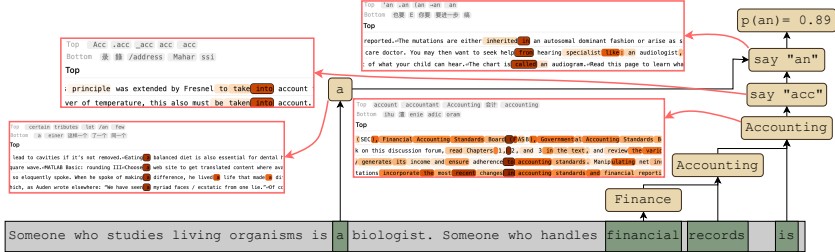

Figure 1: Feature circuit for the input *Someone who studies living organisms is a biologist. Someone who handles financial records is*, explaining Qwen-3 (14B)'s output, *an*. The model plans to say *accountant*, causing it to output the appropriate article, *an*. Labeled nodes are sets of active transcoder features with shared semantics; edges indicate that the source node increases the target node's activation. We demonstrate node semantics by selecting a node's feature and showing its top-activating inputs and the vocabulary items that it up-/down-weights.

with scale. Our results provide the insight into how latent planning emerges at scale, showing that Qwen-3 models use various planning mechanisms that scale with model size. We also show that while both forward and backward planning improve with scale, the former develops faster. We thus conduct the largest-scale feature circuit study on open models to date. We provide code here: `https://github.com/hannamw/model-planning-public`.

## 2 WHAT IS LATENT PLANNING IN LLMS?

Planning is behavior in which one reasons about which actions must be taken (and in which order) to achieve a goal. However, most past work on latent planning in LLMs searches model internals for evidence of a goal, not goal-oriented reasoning. For example, Dong et al. (2025) prompt LLMs to write stories, and probe the LLMs' representations of the input prompt for information about their future outputs. They equate successful probing with latent planning, but see App. G for evidence to the contrary. Pochinkov (2025) takes the residual stream of LLMs that are about to start a new paragraph, and attempts to decode the topic thereof using Patchscopes (Ghandeharioun et al., 2024); again, successful decoding is taken to entail planning. Pal et al. (2023) also decode models' future tokens with probes and Patchscopes—though they do not call this planning. Lindsey et al. (2025) are unique in providing causal evidence: studying LLMs' ability to complete rhyming couplets, they not only observe representations of the rhyming word that the model plans to output, but also causally intervene on them, changing the upcoming word and its preceding context that accommodates it.

We argue that, if LLM planning entails reasoning about the steps needed to output a specific future token, decoding the future token is insufficient to evidentiate planning. Consider a model that always outputs the same token, or one that outputs *0, 2, 4, 6,...*; in both cases, a probe could likely predict many future tokens, but neither task requires planning. More generally, the decodability of a given attribute from model representations does not entail its use in model processing: probes are known to decode unused information (Ravichander et al., 2021). Instead, if latent planning is a *mechanism* that models deploy, a definition thereof should make causally verifiable *mechanistic* claims.

Inspired by Lindsey et al., we define an LLM given a length-$n$ input as engaging in latent planning if it possess a representation of a planned token or concept that:

**Condition 1 (Forward Planning)**: *causes it to output the specific token or concept $t$ at some position $n + k$, $k > 1$.* This strengthens the decodability criterion from past work: we require that some representation *causes* the LLM to produce $t$, not just that $t$ be predictable from the LLM's internals.

**Condition 2 (Backward Planning)**: *causes it to output a context that licenses said token or concept $t$.* This requires that models work backwards from the goal to formulate a context that licenses it. Consider the input $s = $ *The capital of Texas → is → Austin*. LLMs may have an *Austin* representation at the *Texas* position of $s$; ablating it stops the model from later outputting *Austin*. However, this is only backward planning if the *Austin* representation causes the model to produce *is*. This is unlikely, given that one can predict *is* without knowing that *Austin* is the capital of *Texas*. Note that some past work focuses on representations that do *not* aid immediately next-token prediction (Wu et al., 2024).

## 3 TRANSCODERS AND TRANSCODER FEATURE CIRCUITS

To identify causally relevant planning representations, we first decompose model activations into sparse features using transcoders (Dunefsky et al., 2024). Then, we find the causally relevant subgraph thereof, known as a *feature circuit* (Marks et al., 2025; Ameisen et al., 2025).

**Transcoders**  Transcoders are auxiliary models that replace the model's MLPs (Dunefsky et al., 2024); each transcoder takes in one MLP's inputs and predicts its outputs. Formally, a transcoder takes in a given MLP's input activations $\mathbf{h} \in \mathbb{R}^d$ and computes a sparse representation $\mathbf{z} \in \mathbb{R}^n$ as $\mathbf{z} = f\left(\mathbf{W}_{enc}\mathbf{h} + \mathbf{b}_{enc}\right)$. It then reconstructs the MLP's output activations $\mathbf{h}' \in \mathbb{R}^d$ as $\tilde{\mathbf{h}}' = \mathbf{W}_{dec}\mathbf{z} + \mathbf{b}_{dec}$. $f$ is an activation function, while $\mathbf{W}_{enc}, \mathbf{b}_{enc}, \mathbf{W}_{dec}$, and $\mathbf{b}_{dec}$ are learned parameters.

Transcoders are useful because they are trained to compute representations $\mathbf{z}$ that are *sparse* and *monosemantic*: most dimensions (or *features*) are zero on any given input; each feature should fire on only one concept. By contrast, MLPs' hidden activations are often dense and polysemantic, firing on multiple concepts (Olah et al., 2017; Elhage et al., 2022). If one wishes to determine which concepts a model represents in its activations, it is thus easier to interpret transcoder features.

We interpret the $i$th feature of a given transcoder by displaying the inputs that maximize its activation $\mathbf{z}_i$. We also display the tokens whose unembedding vectors have the highest and lowest dot product with the feature's column in $\mathbf{W}_{dec}$; these are the vocabulary items that it directly up- and downweights. See Figure 1 for example feature visualizations, used to manually label features.

We often intervene with respect to transcoder features, to verify our interpretation of a given feature. For example, we might take a feature vector $\mathbf{z}$, set its activation to 0, and observe the change in model behavior. For more background and technical details on transcoders, see Appendix A.1.

**Transcoder Feature Circuits**  Given a model, transcoders trained on each MLP thereof, and an input, we construct a transcoder feature circuit (Ameisen et al., 2025): a weighted acyclic digraph describing the causal relationships between the model's inputs, transcoder features, and logits. Each edge weight indicates the source node's direct effect on the target, i.e. the amount by which it directly increases the latter's value. Once features are annotated, and similar features grouped together, the circuit serves as a mechanistic explanation for a model's behavior on the input, as seen in Figure 1.

We compute feature circuits using Ameisen et al.'s algorithm, detailed in Appendix A.2. Unlike other feature circuit techniques, it computes *exact* direct effect values—conditional on the model's attention patterns and layer normalization denominators. We thus know the precise causal relationship between features, ignoring contributions to these quantities, which is often useful in practice. We use the `circuit-tracer` library for circuit-finding and interventions (Hanna et al., 2025).

The transcoder feature circuit paradigm helps ensure that any planning features found fulfill our conditions, as features are guaranteed to be causally relevant, under the assumptions made by transcoder feature circuits, and we can see what intermediate features represent.

## 4 QWEN-3 MODELS ENGAGE IN SIMPLE PLANNING

### 4.1 MODELS AND DATA

We study planning in 5 models from the Qwen-3 family (0.6B, 1.7B, 4B, 8B, 14B; Yang et al., 2025). We study models of varying size from one family to draw conclusions about how planning behavior develops as models scale. Note that although these models are instruction-tuned, they produce reasonable output on both instruction-formatted and language-modeling-formatted inputs,

| Category | Example Input | Next | Planned |
|---|---|---|---|
| a / an | Someone who handles financial records is | an | accountant |
| is / are | There were 5 dogs but 4 left. Now there | is | 1 |
| el / la | El animal marino con ocho tentáculos es | el | pulpo |

Table 1: Three simple planning tasks. Each task prompts the model to output a **planned** token, preceded by a **next** token with two possible forms; the planned token determines the correct form.

so we use both formats. For feature circuit analyses, we use Hanna et al.'s (2025) transcoders, which cover Qwen-3 (0.6B-14B); we include Qwen-3 (32B) in our transcoder-free behavioral analyses.

We craft three simple tasks to serve as a testbed for LLMs' planning abilities. We choose tasks to which LLMs were likely exposed during pre-training, as model abilities are often stronger on such tasks (McCoy et al., 2024). Each task (Table 1) consists of inputs that push the model to produce a specific content word, preceded by a function word that must agree with it. For example, in the *is / are* task example in Table 1, the model must output 1, preceded by the correct form of *to be*. See App. B for details on the construction and composition of these datasets. We discuss *a/an* in the main text; our successful *is/are* experiments and less successful *el/la* experiments are in App. C / D. For experiments in another language, see experiments with Chinese measure words in Appendix K, while experiments on base models are in Appendix J.

## 4.2 LARGER MODELS SUCCEED ON PLANNING TASKS

We first evaluate models' abilities on the *a/an* task, recording their next token prediction on each input. We report per-class recall, as performance differs by class. We find (Figure 2, left) that all models have high recall ($> 0.8$) of *a*, which is the majority class both in our dataset and English in general. Recall of the minority class *an* is high ($> 0.8$) for Qwen-3 14B; small models (0.6-1.7B) always predict the majority class, and mid-sized models' performance smoothly increases.

Note that this is not attributable to models' inability to determine the planned token: in Appendix E, we show that models with under 14B parameters can calculate the answer to *is/are* questions, but fail to predict the correct verb, producing outputs like ... *there are 1 dog*. It thus appears that simple planning (and not just e.g. math) emerges at 4B to 8B parameters.

## 4.3 MODELS POSSESS PLANNING FEATURES

To determine if models truly plan on these tasks, we compute each model's feature circuit for each example in our datasets, as described in Section 3. We then visualize and qualitatively analyze a subset of the feature circuits, grouping qualitatively similar features together and labeling them.

We find that these circuits contain features that represent the planned token. Figure 1 shows a typical example from Qwen-3 (14B): it possesses planning features (for *accountant*) that feed into features that upweight the same token. These activate features that directly upweight the correct next token (*a/an*). This suggests that models plan to output the target token, which then leads them to output the correct next token. As in Lindsey et al. (2025), the planning features (e.g., the *accounting* feature in Figure 1) appear to simply represent the planned word, and not specifically in planning contexts.

Planning features differ slightly by task. In the *is/are* dataset, such features are more common when the answer is small (from 1 to 3). The *el/la* dataset's features fire on the target word *in English*, despite its lack of grammatical gender, relevant to this task. Surprisingly, on *a/an* and *is/are*, even poorly-performing models have planning features, suggesting nascent latent planning mechanisms.

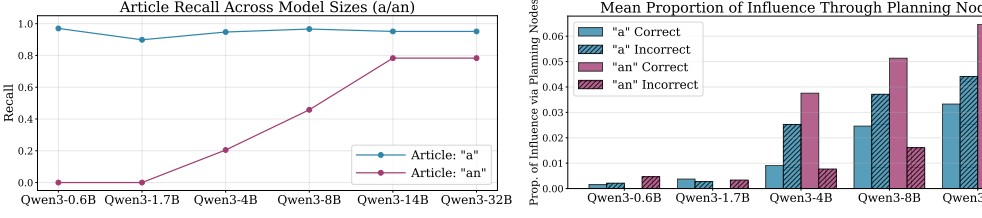

Figure 2: **Left**: Qwen-3 family models' recall of correct article on the *a/an* task. All models can recall *a*, but models $\leq$ 8B have lower recall on the less-common *an*. **Right**: The mean proportion of influence flowing through planning nodes in the *a/an* dataset, by model, article, and correctness. On *an* examples where the model correctly predicts the next token, more influence tends to flow through the planning nodes. This effect is reversed and weaker for the majority class *a*.

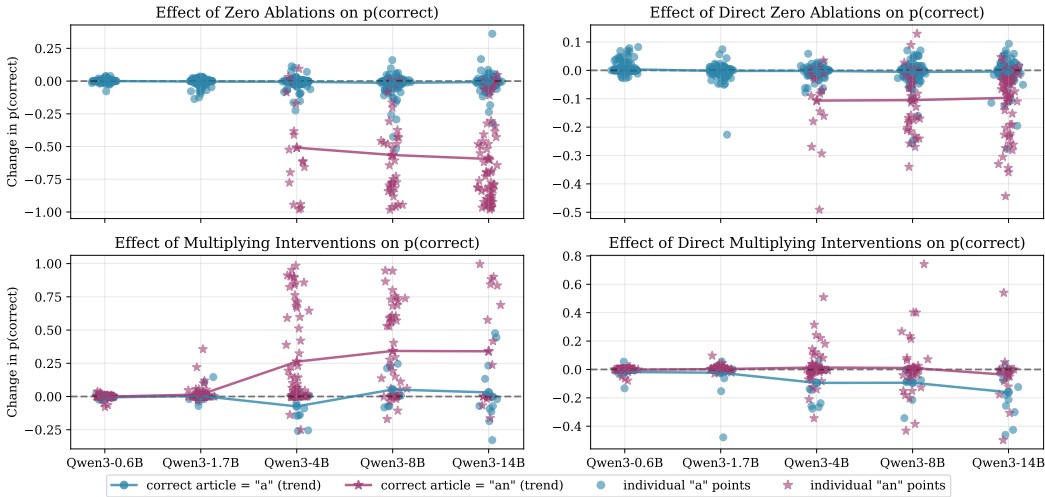

Figure 3: **Left**: Change in $p(correct\ article)$ caused by zero and multiplying interventions on planning features. As expected, ablating these harms performance, while upweighting them improves it; however, both affect primarily *an* examples, the minority class. **Right**: Change in $p(correct\ article)$ caused by direct-effect interventions. Effects are smaller, indicating that planning features act both directly (by upweighting the correct article) and indirectly (by activating e.g *say "a/an"* features.)

### 4.4 PLANNING FEATURES ARE CAUSALLY RELEVANT

We now verify that these planning features truly drive the model's prediction of the correct next token. We start by programmatically finding each example's planning features; a feature is considered planning-relevant if is active at the last position of the input (*is*), and it either upweights the planned word (or a prefix thereof), or contains it in 5 out of 10 of its top-activating texts. We find that this yields similar planning features to those found via manual search.

With these features, we perform two causal relevance analyses. First, we ask—how important are planning nodes according to our circuits? Each edge in the circuit reflects the direct influence of a source node on a target node, but we can also consider the total flow from a source to a target node, which might travel via multi-node paths. To quantify the importance of the planning nodes, we measure the proportion of the total flow between the circuit's inputs and logits that is mediated by the planning features, comparing the flow in cases where the model is in/correct.

We find (Figure 2, right) that when models predict the minority class *an* correctly, more of the total influence flows through the planning nodes. This effect is reversed (and weaker) for the majority-class *a* case, despite roughly equal planning node counts across classes, suggesting that planning nodes are not generally helpful for these examples. In neither case is the proportion large, but this is unsurprising: much of the flow is likely mediated by nodes that identify the need for an article such as *a* or *an*, upweighting them both, rather than discriminating between them.

Second, we causally intervene on planning features. For each model, we a) take the examples on which it succeeds and ablate the planning features (e.g. *accountant* and *say "acc"* in Figure 1), setting them to zero, and b) take the examples on which it fails and highly upweight their planning features, setting their activations to $5\times$ their usual activations. If these features indeed cause models to output the planned token, these interventions should harm and improve performance respectively.

We find (Figure 3, left) that features are indeed causally relevant. Feature ablation (top left) harms model performance, but only on minority-class *an* examples. Similarly, boosting planning features improves performance drastically *an* examples, with larger models seeing slightly larger improvements; however, the effects on *a* examples are almost zero. This asymmetry aligns with our prior analysis, and suggests that planning nodes are more important for minority-class examples where models must work against their priors. This intervention is effective on Qwen-3 4B and 8B, indicating that although their overall performance is worse than Qwen-3 14B, they likely rely on similar planning mechanisms, with planning features encouraging the production of *an* when necessary.

We also note that our feature interventions are more successful than a random baseline: while zero ablating randomly selected features active at the last position of the prompt occasionally harms performance, multiplying random features fails to boost model performance (see App. F for details).

**Discussion**    Our results suggest that Qwen-3 engages in simple backward planning; however, it is unclear if this is driven by direct-effects alone. The *accountant* feature might have a high cosine similarity with the unembedding vector for *an*, upweighting its logit. This, combined with a mechanism that upweights both *a* and *an* in relevant contexts, would suffice to upweight the correct article, as we observe. We disprove this by performing direct-effects interventions: we upweight the planning features, but freeze the model's other features, blocking second-order effects.

This intervention's effects (Figure 3, right) are much weaker than the original interventions: zero ablations are less harmful, and multiplying interventions harm performance as often as they help. The planning features' importance can thus not be explained by direct effects alone, suggesting that the *say "a/an"* features play an important role in mediating planning. For more evidence, see Appendix L, where we steer on the planning features, and find that while this often causes models to output the planned-for word, it seldom causes them to output *a/an*.

One could also hypothesize that although *say "a/an"* features are involved in *a/an* planning, the model treats noun phrases (like *an accountant*) like a single, multi-token word; no planning is involved. However, models also plan when outputting "there *is 1* dog left", where this multi-token argument is much less plausible. We thus maintain that simple planning occurs in these cases.

One outstanding question is the source of the gap between Qwen-3 (14B) and its weaker 4B and 8B counterparts; what makes their circuits *nascent* rather than fully-developed? In Appendix M, we examine this question and find that when these mid-sized models fail on *an* examples, they have far fewer planning features active than when they succeed. By contrast, smaller models seldom have any planning features active, and Qwen-3 (14B) has many planning features active in both cases.

## 5    QWEN-3 USE LITTLE PLANNING WHEN COMPLETING COUPLETS

The preceding experiments show that Qwen-3 models more successfully plan as their size increases, but leave open the question of longer-range planning mechanisms. There is precedent: Lindsey et al. (2025) find that, given the first line of a rhyming couplet, like *He saw a carrot and had to grab it,* Claude-3.5 Haiku produces the next line *His hunger was like a starving rabbit* using a *rabbit* feature that controls the rhyming word and generates a coherent context. Motivated by this, we study Qwen-3 models on rhyming couplets, searching for long-range planning.

### 5.1    QWEN-3 MODELS OFTEN SUCCESSFULLY RHYME COUPLETS

We first test whether Qwen-3 models can complete rhyming couplets at all. To do so, we generate a dataset of 985 first lines of couplets, by prompting Qwen-3 (32B) to produce rhyming couplets on 43 topics, ranging from *coming of age* to *animals and wildlife*, and taking the first line of each. LLM generation of couplets avoids cases of couplets memorized from the training data. We then greedily sample a second line of the couplet from each model, and evaluate its rhyme with the first couplet by extracting the last word of each line, extracting their vowels and final consonants using CMUDict (Carnegie Mellon University, 2014; Bird & Loper, 2004), and verifying that they match. Our results (Figure 5, left) show that larger models rhyme with 50+% accuracy; smaller ones fail more often. Models engage in slant or assonant (vowel-only) rhyme, rhyming words like *craze* with *page*; models with 8B parameters produce a valid assonant rhyme in over 70% of cases.

### 5.2    LARGER LLMS' POETRY ABILITIES ARE SUPPORTED BY A RHYMING CIRCUIT

To test whether models plan when completing couplets, we again use transcoder circuits. For each model, we filter the examples from our dataset to those where the model completes the couplet's second line with a rhyming word. We then attribute from this rhyming word's logit, given the input leading up to the rhyming word; that is, given an input like *Fury burns where calm once stayed,...Hope flickers where the shadows laid*, we find the circuit explaining the model's prediction of *laid*. We limit this to 100 examples per model. See App. H.1 for rhyming couplet data details.

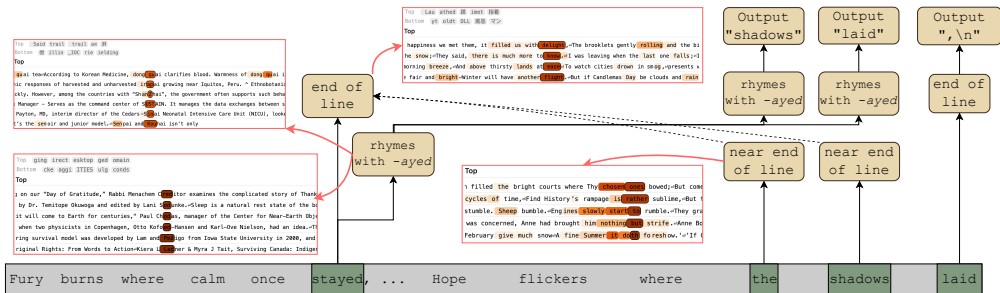

Figure 4: A feature circuit for the couplet *Fury burns where calm once stayed,\n Hope flickers where the shadows laid*, explaining Qwen-3 (14B)'s decision to output *shadows laid,\n*. Halfway through outputting the couplet's second line, the model's "near end of a line of poetry" features activate. These cause it to attend back to the end of the first line, where "end of a line of poetry" features are active, and to move "rhymes with -*ayed*" features into the second line. These influence the model's outputs, eventually leading to *laid*. *End of line* features then cause it to output .\n.

We qualitatively analyze the circuits, and find that in larger models, an interpretable circuit emerges. Given the first line of the couplet, the model begins to generate the second with little planning. Near the end of the second line, the model recognizes that it is near the end of a line of rhyming poetry, activating *near end of line* features. These cause it to attend to the end of the first line, drawn by the *end of line* features active there. Rhyming features (e.g. *rhymes with "-ayed"*) at the end of the first line thereby activate similar features in the second line. There, these features remain active until they eventually cause the model to output a rhyming token. Once the model completes the rhyme, it activates *end of line* features and stops generation. Figure 4 depicts this process.

We defer detailed evidence for our circuit to Appendix H.2. There, we show that *end of line* features are causally responsible for both the model's decision to end a line of poetry, and for indicating where the model should attend to, in order to extract the rhyming features. We similarly show that the *near end of line* features cause the model to attend back to the *end of line* feature position. Here, we focus on the question: does the couplet circuit involve planning?

## 5.3 QWEN-3 MODELS PLAN FORWARD, BUT NOT BACKWARD, TO COMPLETE COUPLETS

If the couplet circuit involves planning, we view the rhyming features at the end of the couplet's first line as the most likely planning features. They clearly represent the rhyme to be output, and our circuits indicate that they influence the model's decision to output rhyming words. However, we must still test that both forward and backward planning occur when models generate rhymes.

We first define rules to automatically find rhyming features. This is challenging, as Qwen-3 models represent e.g. an -*ayed* feature via separate -*ai*- and -*d* sound features, which specify the vowel and final consonant of the rhyme. The top-activating tokens for such features tend to be subwords, and may employ multiple, potentially nonstandard spellings for a given sound; see Figure 4 for example features. As a heuristic, we identify features whose top-10 max-activating tokens are short (under 5 characters), and do not represent a single word (they activate on the same word at most 5 times). We also require that at least 7 of these 10 tokens start with the same vowel, or end with the same consonant, to ensure that the feature's top-tokens all represent one sound. This definition captures rhyming-relevant features with relatively high precision but only moderate recall.

Next, for each couplet, we downweight the rhyme features at the end of its first line, multiplying their activations by -3. We then sample a random couplet with a distinct rhyming sound, and upweight its rhyme features, multiplying their original activations by 7; we find these steering hyperparameters via manual search. We then generate a completion to the first line of the original couplet, while steering on the end of the first line. We measure rhyming accuracy with respect to the new rhyme.

It is harder to quantify Condition 2—whether a given context licenses a specific word (or set thereof) as opposed to licensing many words. However, we can test which context (the original or steered one) best enables the model to predict the new rhyme, when the model is steered towards the new rhyme. If the new context indeed licenses the new rhyme better, the model should more accurately

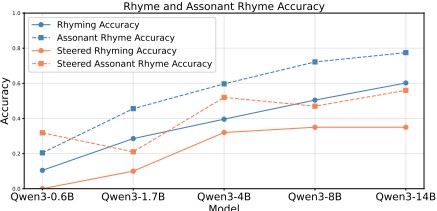 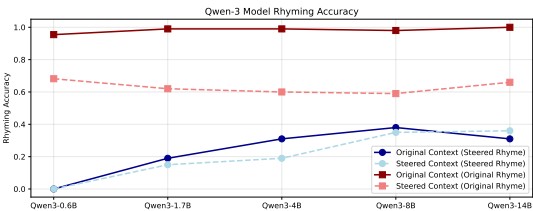

Figure 5: **Left**: Qwen-3 rhyming accuracy. In the base case, models have moderate rhyming accuracy, reaching 0.6 at 14B parameters (solid blue); when we consider assonant (vowel-only) rhyme, Qwen-3 (14B) achieves 0.8 (dashed blue). When steered to predict a new rhyme, model accuracy is only moderate for perfect rhymes (solid orange), but improves with scale, and is better on assonant rhyme (dashed orange). **Right**: Model rhyming accuracy when trying to predict a token satisfying the couplet's the steered rhyme (blue lines) or original rhyme (red lines), given the original (solid) or steered (dashed) context. The model predicts the steered-for rhyme with similar accuracy given the original or steered context. This suggests that the steered context does not better license the rhyme.

predict a rhyming word given it. To test this, we feed the model both the original and steered couplet completions, with their last word removed. We then record the model's generation given each, when steered towards the new rhyme, and compute rhyming accuracy with respect to the new rhyme.

We find that models do engage in forward planning: Figure 5 (left) shows that steering on the rhyme features does change the model's rhyme to the new rhyme in the case of larger models (8B-14B). Though accuracy is only moderate (40%), normal rhyming accuracy was similarly modest at 60%, and assonant rhyme accuracy is higher (up to 60%). Moreover, we observe that steering changes both the final rhyming word and the intermediate context; see Appendix H.3 for quantitative evidence.

However, Figure 5 (right) shows that the intermediate context generated under intervention does not necessarily license the new rhyme better. When we steer the model, it is equally likely to output the injected rhyme given the steered intermediate context (light blue, dashed line) as when given the original one (dark blue, solid line). Giving the model the intermediate context produced with steering, but not steering it, elicits the original rhyme with relatively high accuracy: near 60% across models (light red, dashed line). This is low compared to the accuracy given the original context (near 100%; dark red, solid line), which could suggest that the original context better licenses the original rhyme. However, the fact that we only intervened on examples where models rhymed successfully inflates this accuracy. Overall, these results suggest a lack of strong backward planning.

## 5.4 LARGER MODELS MAY USE LOCAL PLANNING FEATURES

Though the backward planning results are mostly negative, results for larger models (8B-14B) trend in the right direction: they more accurately predict the steered rhyme given the steered context, and less accurately predict the original rhyme; in App. H, we see that their steered generations overlap less with original generations. Moreover, manually inspecting Qwen-3 (14B) couplet-completion circuits showed that while most couplet circuits involve second-line rhyming features that upweight rhyming words, some instead involve *say X* features that upweight a specific upcoming word. These often coincide with rhymes that require some setup, such as a *say "night"* feature occurring before the model outputs *in the night*. These are prime candidates for *local* planning features, that plan for short phrases, but not whole lines; we thus test whether they elicit backward planning in models.

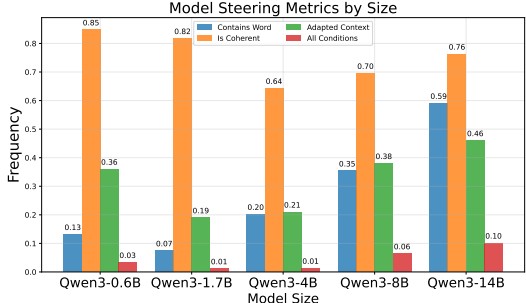

Figure 6: Adaptation metrics by model. As models grow, so does the proportion of (1) outputs containing *X* (blue), and (3) coherent and *X*-containing outputs that also adapt the context to license *X* (green). Few examples do all three.

We first identify potential planning features, searching our couplet circuits for *say X* features that upweight the output rhyming word, but are active prior to when *X* is output: such features might adapt the preceding context to license that word. We then steer models using these features on 100 inputs from the TinyStories dataset (Eldan & Li, 2023), which we use as a source of neutral input text. For each steered output, we check if it (1) contains the steered word, (2) is coherent, and (3) adapted the context to fit the steered word. We evaluate (1) programmatically, use Claude 4 Sonnet to evaluate (2), and manually verify (2) and evaluate (3) on a subset of outputs that satisfy (1) and (2). See Appendix I for experimental details.

We find (Figure 6) that steering on these *say X* features often induces models to output *X* (blue bars). Moreover, for outputs that are coherent and contain *X*, models—especially larger ones—do adapt their outputs to produce whole phrases like *in the night* or *had a recurring dream* (green bars). The scaling trend likely occurs because larger models have more such local planning features in their couplet circuits. However, this phenomenon is sensitive to steering strength, and these features occur only in a small minority of couplets. We hypothesize that such features are part of an emerging planning mechanism in larger models, much as *a/an* and *is/are* planning can be seen to emerge in Qwen-3 (4B); at larger scale, models may more reliably engage in local planning. Still, more study is needed to confirm the role these features play.

## 6    RELATED WORK

**(Feature) Circuits**    We build on prior work on circuits, which attempts to capture an (ideally minimal) set of units that are causally relevant to and explain a model's behavior on a task (Olah et al., 2020; Elhage et al., 2021; Conmy et al., 2023). Early LLM circuits were composed of attention heads and MLPs, and explained how models performed indirect object identification and the greater-than operation (Wang et al., 2023; Hanna et al., 2023). Circuits composed of features from sparse autoencoders or transcoders have the added benefit of having interpretable nodes; however, finding them is expensive and requires auxiliary models. They have been used to explain gender bias, syntactic processing, and more (Marks et al., 2025; Hanna & Mueller, 2025; Lindsey et al., 2025).

**Planning Tasks**    LLMs' grammatical agreement abilities, as in our **a/an**, **is/are**, and **el/la** tasks, have been widely studied. LLMs generally excel at agreement, preferring sentences with correct agreement over incorrect ones (Warstadt et al., 2020; Chang & Bergen, 2024). Prior mechanistic work on agreement is more limited to *is/are* and the broader phenomenon of subject-verb agreement: past work has found linear subspaces, neurons, and sparse features relevant to it (Lasri et al., 2022; Finlayson et al., 2021; Brinkmann et al., 2025). Past work has studied LLM **poetry and rhyming abilities** in the context of building and evaluating poem-generating systems (Sawicki et al., 2023; Chen et al., 2024; Suvarna et al., 2024); Lindsey et al. (2025) provide the first mechanistic study.

**Planning Mechanisms**    Section 2 discusses past work, but contemporaneous work also addresses planning: Nainani et al. (2025) search for code planning feature circuits in Gemma-2 (2B; Gemma Team, 2024), while Maar et al. (2025) investigate poetry abilities across models using probes.

## 7    DISCUSSION

**How General Are the Discovered Planning Mechanisms?**    Our evidence suggests that the planning mechanisms we discover are not general in the sense that the model uses the exact same circuit for all planning tasks. Whether a given model plans on a given task is regulated by the model's capacity, as well as the task's complexity, along with its frequency and importance (in terms of training loss). Thus, larger models plan more, and common planning tasks are learned faster, resulting in piecemeal planning abilities, rather than a large set of abilities and a unified mechanism.

Successful planning circuits often follow a motif: there are planning features indicating the planned word, which then activate downstream features responsible for backward planning. However, models may learn to plan in one case (*a/an* agreement) but not others (couplets), simply because the former is more important to reducing its loss than the latter, and the model is too small to learn both.

**Extensions to Complex Tasks** In this paper, we study relatively simple tasks, as Qwen-3 (14B) and smaller struggle with complex planning tasks (like e.g. chain-of-thought unfaithfulness, as in Lindsey et al. (2025)); this prevents us from studying such tasks. However, as open source models become more competent, circuit-tracing should still be able to address these planning behaviors. Sometimes, as in chain-of-thought unfaithfulness examples where the model plans for a single-token answer, it may suffice to simply attribute from the given answer token, as done here.

In other cases, e.g. detecting a hidden goal that drives model behavior without producing one "smoking gun" answer token, we may want to attribute back from more general high-level model actions, such as "Why did the model produce a refusal?" or "Why did the model make a given suggestion?". Though little past work does so, attributing from such higher-level actions is possible by attributing from arbitrary directions in activation space. In this case, one must identify a direction in activation space corresponding to such an action, and attribute from this. One could identify causally relevant directions for such actions via probing or difference-in-means approaches.

**Cross-model Generalization** In this paper, we study only one model family, to focus the effects of scale on planning. The lack of transcoders for similarly competent models currently hinders comparisons across model families, but we believe such cross-model comparisons would be valuable. That said, past work indicates that circuits can generalize across model family and scale: the circuits for e.g. multi-hop reasoning in Gemma-2 (2b) (Hanna et al., 2025) look similar to those in Claude (Lindsey et al., 2025), a vastly larger and more competent model.

We also note that our planning framework can be applied without transcoders. For example, one could train probes to extract a planned word or rhyme family and perform interventions with respect to them; see Maar et al. (2025) for work along these lines. However, this approach is less flexible, requiring new probes for each task, and losing the fine-grained insights of transcoder feature circuits. Recent work suggests that circuit-tracing may even be possible with neurons alone (Arora et al., 2025), allowing for both fine-grained insights and cross-model comparison.

## 8 CONCLUSION

Our experiments have shown that some Qwen-3 models engage in latent planning, possessing features that represent the planned word and causally influence both the output word and the context preceding it. Both forward and backward planning abilities improve with scale, but the former improves before the latter; even in Qwen-3 (14B), planning multiple tokens ahead is rare.

Why might planning only begin to emerge at scale? We hypothesize that planning, especially backward planning, is costly to implement: models must learn not only to plan for a specific token, but also how to plan backwards for it in a context-specific way; *a/an* planning and couplet planning have distinct mechanisms. Thus, models may learn to plan only after exhausting other, simpler ways of reducing their loss. Bachmann & Nagarajan (2024) suggest that teacher forcing in LLM pre-training may also reduce the pressure to plan: even if a model fails to backwards-plan for a crucial agreeing token like *an*, teacher forcing provides that token anyway. Models that suffer the consequences of their poor planning, such as those trained with on-policy reinforcement learning methods, may thus face more pressure to plan.

Whatever the reasons behind this, latent planning abilities in Qwen-3 models up to 14B parameters are still nascent. This is relevant for scheming, an AI safety risk where models work towards secret goals (Balesni et al., 2024); past work has induced scheming in models, and caught them by reading their chains of thought (Meinke et al., 2025). Models with strong latent planning abilities might thus cause concern, but we observe little complex planning in Qwen-3. What we observe instead is latent planning abilities that appear to improve with scale—and merit monitoring as models grow.

In this paper, we have provided a framework for doing precisely that; however, monitoring latent planning with feature circuits is still a significant technical challenge. Open-source circuits work on models above 8B parameters is still rare. Large-scale work on feature circuits is yet scarcer, due to the lack of open transcoders for large models. As mechanistic interpretability's seeks interpret more sophisticated behaviors, its methods must scale to match the models that possess them.

## REPRODUCIBILITY STATEMENT

We conduct our experiments with openly available models, including both LLMs and transcoders. We release the data and code used as part of this study in the following repository: `https://github.com/hannamw/model-planning-public`. Our experiments can be run with as little as 40GB of GPU RAM, though they will run much faster on 80GB of memory, and quite quickly (around 1 GPU-day) on 140GB of RAM (i.e. one NVIDIA H200 GPU). Note that features, models, transcoders, and circuits are large; we recommend having 3TB of disk space available.

### ACKNOWLEDGMENTS

The authors thank Denis Paperno and Jim Marr for insightful conversations, and thank the Anthropic Fellows Program for enabling and supporting this project. MH is supported by an OpenAI Superalignment Grant and a Google PhD Fellowship.

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

# A  DETAILS OF TRANSCODERS AND FEATURE CIRCUITS

## A.1  TRANSCODERS

In this section we provide details on transcoders in general and the specific transcoders we use.

**Transcoders**  Past work has attempted to characterize the features encoded in model activations by examining the inputs that most strongly activate each neuron (dimension) of a given activation vector. However, interpreting neurons is difficult, as they are seldom zero and often polysemantic, firing for multiple reasons (Olah et al., 2017; Elhage et al., 2022). Sparse dictionary learning solves this problem by decomposing activations into sparse and (ideally) monosemantic feature vectors (Olshausen & Field, 1997; Bricken et al., 2023). As only a few dimensions, or *features*, of the vector are active on a given input, and each feature fires on only one concept, these are much easier to interpret.

Sparse dictionaries come in many forms. Sparse autoencoders (SAEs; Bricken et al., 2023; Huben et al., 2024) are the most common type, encoding and reconstructing activations from the same location. We use *per-layer* transcoders, which encode MLP inputs and reconstruct MLP outputs (Dunefsky et al., 2024); see Figure 7 for a diagram. Lindsey et al. (2024) also introduce *cross-layer* transcoders, which take in MLP inputs, and are jointly trained to predict contributions to all downstream MLPs' outputs. These are generally sparser (for a given level of reconstruction error) but also more computationally costly to train and more memory-intensive to deploy. Importantly, while Ameisen et al. (2025) use cross-layer transcoders for their circuit-finding, per-layer transcoders can also be used.

Formally, a (per-layer) transcoder takes in activations $\mathbf{h} \in \mathbb{R}^d$ from a given MLP's inputs, computes the sparse representation $\mathbf{z} \in \mathbb{R}^n$, and reconstructs the MLP's output activations $\mathbf{h}' \in \mathbb{R}^d$ as follows:

$$\mathbf{z} = f\left(\mathbf{W}_{enc}\mathbf{h} + \mathbf{b}_{enc}\right) \tag{1}$$

$$\tilde{\mathbf{h}}' = \mathbf{W}_{dec}\mathbf{z} + \mathbf{b}_{dec}, \tag{2}$$

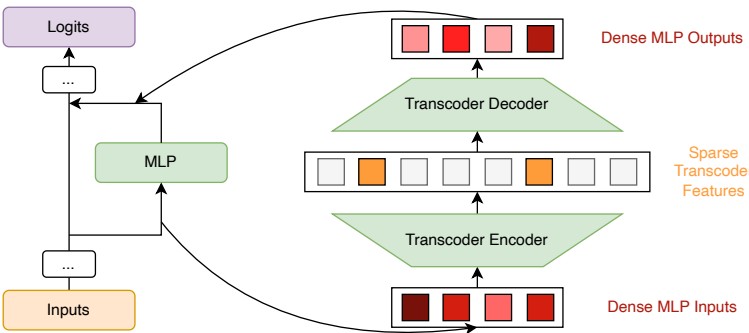

Figure 7: A diagram of a transcoder. The transcoder takes in the dense MLP inputs, computes a sparse representation thereof, and then reconstructs the MLP's dense outputs.

Here, $f$ is an activation function (often ReLU, JumpReLU, or Top-$k$), and $\mathbf{W}_{enc}, \mathbf{b}_{enc}, \mathbf{W}_{dec}$, and $\mathbf{b}_{dec}$ are learned parameters. LLM transcoders are trained to minimize both the MSE between $\mathbf{h}'$ and $\tilde{\mathbf{h}}'$ and the norm of $\mathbf{z}^2$. The LLM is frozen, and the transcoder trains on up to billions of tokens. The reduction in polysemanticity is achieved by setting the sparse representation size to be much larger than the input size. In doing so, one reduces the pressure on the model to cram many features into a small number of dimensions, as is thought to cause polysemanticity (Elhage et al., 2022).

We interpret the $i$th feature of a given transcoder by displaying the inputs that maximize its activation $\mathbf{z}_i$. We also display the tokens whose unembedding vectors have the highest and lowest dot product with the feature's column in $\mathbf{W}_{dec}$; these are the vocabulary items that it directly up- and downweights. See Figure 1 for example feature visualizations, used to manually label features.

We often intervene with respect to transcoder features, to verify our interpretation of a given feature. To do so, we take the original feature vector $\mathbf{z}$ and perform desired interventions on it by e.g. zeroing out a feature's activation, yielding $\mathbf{z}'$. We compute $\Delta = \mathbf{W}_{dec} (\mathbf{z}' - \mathbf{z})$, and add $\Delta$ to the output of the corresponding MLP during the model's forward pass.

**Qwen-3 Transcoders**  For our experiments, we use Hanna et al.'s (2025) Qwen-3 transcoders. These circuits are ReLU transcoders, all with a hidden dimension of 163840. They take in MLP inputs post-input-LayerNorm, and predict the MLP's outputs.

## A.2 Transcoder Feature Circuits

Formally, feature circuits are weighted acyclic digraphs. The source nodes are input embeddings and nodes corresponding to each transcoder's reconstruction error $\tilde{\mathbf{h}}' - \mathbf{h}'$. These flow through transcoder feature nodes, to nodes that correspond to a given vocabulary item's logit. Each edge's weight is the direct effect of the source node on the target, i.e. the source node's effect on the target's value, unmediated by other nodes.

We compute feature circuits using Ameisen et al.'s (2025) algorithm, which works as follows.

**Local Replacement Model**  The first step of attribution is to incorporate the transcoders into the model's computations for a given input. We thus replace the model's MLPs with their corresponding transcoders, plus a reconstruction error term equal to the difference between the MLP's output and the transcoder's reconstruction. This yields a *local replacement model*, which behaves identically to the original model, but only on the given input, as reconstruction error terms are input-specific.

We next freeze the model's attention patterns and denominators of any layer normalization terms, treating them as constant values; this entails detaching them from the graph (`.detach()` in Pytorch). See Figure 8 for a depiction of this process. We also detach the transcoder feature activations themselves, so no gradients flow through them.

---

[2]This is often done by penalizing $\mathbf{z}$'s $L_1$ norm. However, note that some activation functions, namely Top-$k$ and variants, inherently limit the number of active features, making this unnecessary.

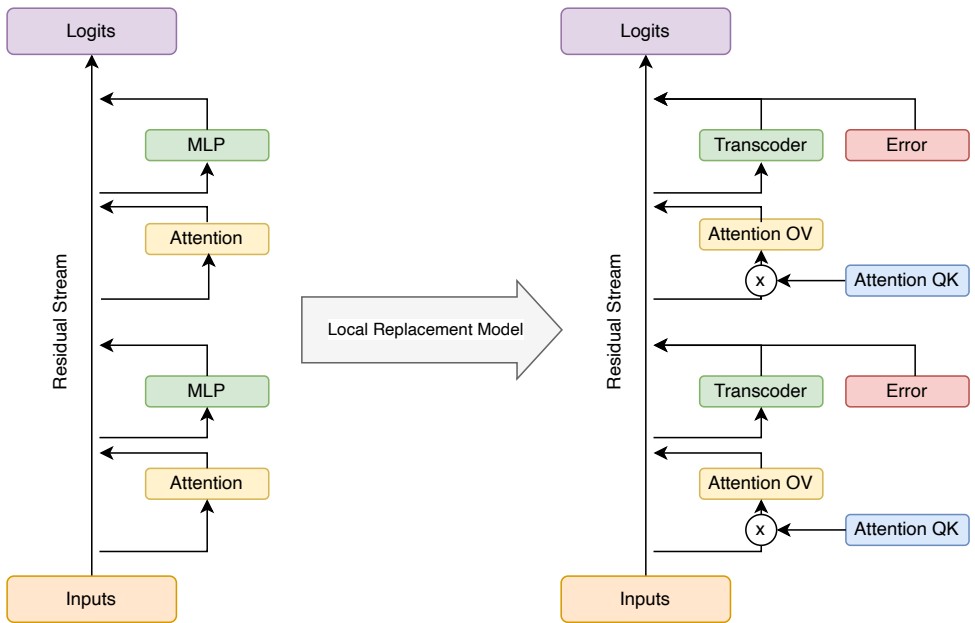

Figure 8: A 2-layer transformer LM, and its corresponding local replacement model. We replace model's MLPs with transcoders, as well as error terms unique to the given input. The attention patterns (from the QK matrix) have been frozen, detaching them from the computation graph. Despite this, the OV-matrix of each attention block is still attached. Thus, when we refer to e.g. the direct effect of a feature of the layer-0 transcoder on a vocabulary logit, this direct effect may pass through the residual stream alone, or additionally through the OV matrix of the attention, a linear transformation. The direct effect of any given feature on any other feature (or any vocabulary logit) is thus linear. See Elhage et al. (2021) for more on QK/OV matrices and the residual stream.

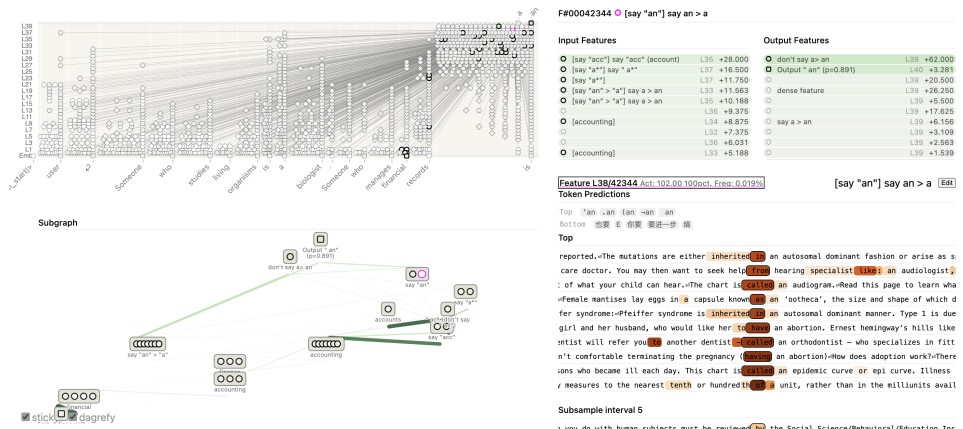

Figure 9: The interface used for circuit visualization / annotation, from `circuit-tracer`.

In so detaching these components, we remove all nonlinearities from our local replacement model: both the attention softmax nonlinearity and the normalization nonlinearities are gone. The activation of any given feature is now linear in the activations of the nodes prior to it. This simplifies the process of computing the direct effect of one node on another, and means that these direct effect values are exact; however, they will not account for features' impact on the model's attention *patterns*.

**Attribution** We can thus compute the direct effects of a source node on a target node as follows. We define an input vector for the target node: if the node is a feature, this is its input vector (from $\mathbf{W}_{enc}$), and if the node is a logit, this is the corresponding unembedding vector, minus the mean unembedding vector. We inject this gradient at the node's input location—either the MLP input for transcoder features, or the final residual stream for logit nodes; this injection can also be operational-ized as a dot product with the residual stream, followed by a `.backward()` call. Then, for each upstream node, its direct effect is the gradient at its output location, multiplied by its output vector: the input embedding or error vector for input and error nodes respectively, or the source feature's activation multiplied by its decoder vector, for feature nodes. With each call of `.backward()`, we find weights for all edges into the target node; repeating this for all nodes attributes the whole attribution graph.

**Methods** We limit attribution to the top 7500 most influential feature nodes, as remaining nodes are unlikely to be important, and attributing from many nodes leads to large graphs that fit poorly in memory. We determine which nodes are most influential by intermittently computing each node's influence using the procedure described in Ameisen et al. (2025). For logit nodes, we choose to attribute from the minimum required to capture 0.95 of the model's next-token probability, or the top 10 logit nodes, whichever is smaller (generally the former). Ultimately, the attribution process is quick, from seconds for Qwen-3 (0.6B) to a minute or two for Qwen-3 (14B).

For visualization purposes, it is often useful to prune graphs, removing low-influence nodes and edges. As done by Ameisen et al. (2025), we do so by computing the total influence of each node and edge in the circuit. We then set a threshold for each, and take the minimum number of top nodes / edges that sum to that influence; we choose nodes whose influence sums to 80% of the total, and edges whose influence sums to 98%. Our circuit-finding interface, provided by `circuit-tracer` (Hanna et al., 2025), is shown in Figure 9.

## B Simple Planning Dataset Details

We construct three datasets for testing simple planning, the *a/an*, *is/are*, and *el/la* datasets. The *a/an* dataset consists of 108 examples of professions (86 requiring *a* and 22 requiring *an*) and descriptions thereof. These were augmented with 350 concrete nouns (267 *a* / 83 *an*) and descriptions thereof. All descriptions were generated by Claude 4 Sonnet, but filtered manually and rewritten if necessary, e.g. because they were too vague. The *is/are* dataset was generated programmatically, and consists

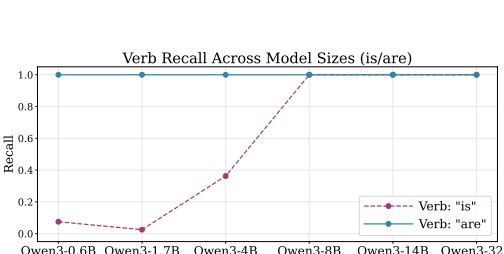 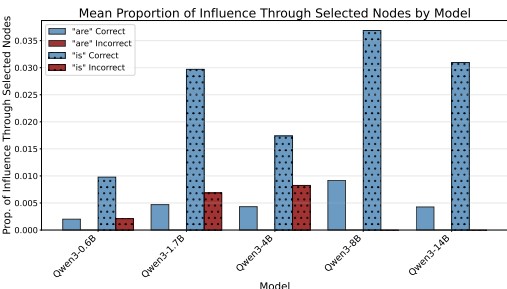

Figure 10: **Left**: Recall of *is* and *are* on the *is/are* dataset, by model. Models below 8B in size mostly fail to predict *is*, while larger models perform perfectly. All models can predict *are*. **Right**: The mean proportion of influence flowing through planning nodes in the *is/are* dataset, by model, verb, and correctness; recall that the only incorrect examples are small models failing to predict *is*. The most influence flows through the planning nodes in the *is* examples, where more planning nodes are present. Still, more influence flows through these nodes in correct than incorrect *is* examples.

of (positive) differences between numbers ranging from 1 and 9; the animals are sampled from a manually curated list of 10 animals. This yields 360 examples. The *el/la* dataset, much like *a/an*, consists of 411 concrete nouns (223 *el* / 188 *la*) and Spanish-language descriptions thereof. Again, all descriptions were generated by Claude 4 Sonnet, but filtered manually and rewritten if necessary.

Note that, in the case of the *a/an* dataset, one randomly-sampled in-context example from our dataset is prepended to each input to the model in order to encourage it to output *a/an*; otherwise, the model does not understand the task structure, and outputs other tokens. The full prompt is thus something like `Someone who provides treatment for physical or mental conditions is a therapist. Someone who heals sick pets is.` This is fed directly to the model as the user input, and the model simply completes the input (rather than generating a separate assistant response). The *el/la* dataset is formatted in the same way.

Similarly, we prepend *is/are* examples with *Repeat and finish the following sentence:*, as we found that this increased performance over simply sampling next tokens without requesting the repetition. The full prompt is thus something like `/no_think Repeat this sentence and complete it. At first there were 2 cats. Then, 1 went away. Now, there.` The `/no_think` prevents models from thinking before answering. During attribution, we prefill the model's assistant response with `<think>\n\n</think>\n At first there were 2 cats. Then, 1 went away. Now, there.` We then attribute back from the top logits (which are always *is* and/or *are*).

## C *Is-Are* RESULTS

Here, we report results for experiments on the *is/are* dataset, which largely mirror those performed on the *a/an* dataset. Figure 10 (left) shows that models behave similarly on the *is/are* to the *is/are* dataset: all models do well on the majority class *are*. Models below 8B in size fail (0.6-1.7B) or perform poorly on the task when the correct answer is *is*; Qwen3-4B scores just below chance. Starting at 8B, models score perfectly on *is* as well, just as with *a/an*.

We perform circuit analysis on *is/are* dataset as well, and find similar, but not identical trends compared to the *a/an* case. Models again have features corresponding to planning features some of the time. However, *1* features (and *2* and *3* features to a lesser extent) are more common than other numbers' features. Whether this is a real phenomenon (models have special representations for lower numbers due to their frequency) or a transcoder-driven phenomenon (higher numbers also have corresponding features, but transcoders miss these) is unclear. This may also be related to the fact that such features are more important / necessary in the minority class case (*1/is*) than in the majority class case. In the case where such features do exist, we also observe that e.g. *1* features activate features that induce the model to say *is*.

We perform the flow and intervention experiments done on the *a/an* dataset. These are complicated somewhat by the fact that there are more planning nodes in the *is* case than in any of the *are* cases,

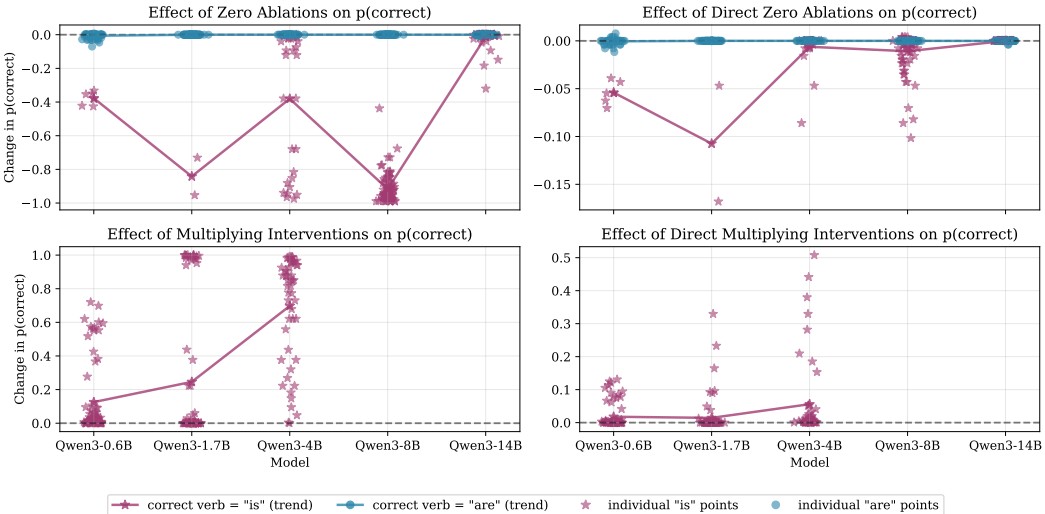

Figure 11: **Left**: Change in $p(correct\ verb)$ caused by zero and multiplying interventions on planning features. The former generally harm performance, while the latter improve it. Both affect only *is* examples, which have the most planning nodes, and also are the only examples models answer incorrectly. **Right**: Change in $p(correct\ verb)$ caused by *direct* zero and multiplying interventions on planning features. As before, these are relatively ineffective, though less so than in the *a/an* case.

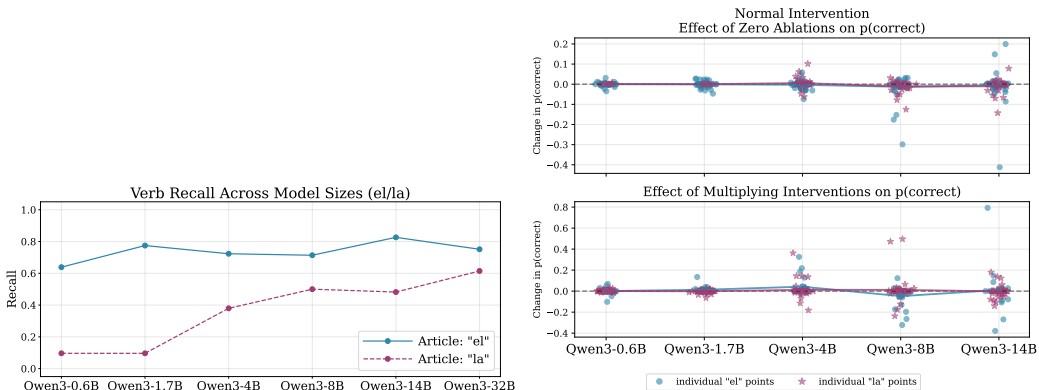

Figure 12: **Left**: Recall of *el* and *la* examples by Qwen-3 models. Unlike in prior examples, the majority class (*el*) is not perfectly captured by any model, though recall is generally high. Moreover, while performance on the minority class *la* improves with scale, recall is ultimately still middling. **Right**: Interventions performed with respect to *el/la* planning features fail primarily due to a lack of said planning features.

and that models do not fail on *are* cases. Still, in Figure 10 (right), we can see that in the *is* case, more influence flows through the planning nodes in correct than incorrect examples, as in the *a/an* dataset. Moreover, Figure 11 (right) shows that both zeroing and multiplicative interventions are effective *on is examples*. This is likely because these have the most planning nodes; however, it may also be related to the fact that *is* is the minority class, and "needs" these features more.

# D  *El-La* RESULTS

Here, we report results for experiments on the *is/are* dataset, which are much less successful than those performed on the *a/an* or *is/are* datasets. When we behaviorally test the models on this task, we find (Figure 12, left) that performance is worse than on the prior two tasks. The majority class *el* is not always correctly predicted, though performance stays steadily high as in other tasks. Moreover,

while recall of the minority class *an* does improve with model scale, it never exceeds 0.6, unlike on other tasks, where it reaches near 1.0.

We then perform the causal interventions, using as planning nodes those that either in Spanish or in English, as we observe that some examples have English nodes corresponding to the hypothetically planned word. However, we find (Figure 12, right) that the interventions have little effect; this goes for both zero and multiplying interventions.

We believe that this is primarily driven by a lack of planning features active on these examples. In general, while we can find some planning features, Qwen-3 models simply have much fewer than they do on the *a/an* dataset, despite their formats being very similar. This may be because Qwen-3 is not highly capable in language besides English and Chinese (which exhibits little syntactic agreement); further studies could examine more multilingually capable models.

## E    MODEL PERFORMANCE ON NON-PLANNING ASPECTS OF SIMPLE PLANNING TASKS

The fact that small models fail to plan on the simple *a/an* and *is/are* planning tasks may raise the question: do small models fail because they cannot perform the tasks at all? To show this is not the case, we generate models' planned tokens, both given the correct next token, and the incorrect next token. We then measure whether the output token in each case matches our expected planned token.

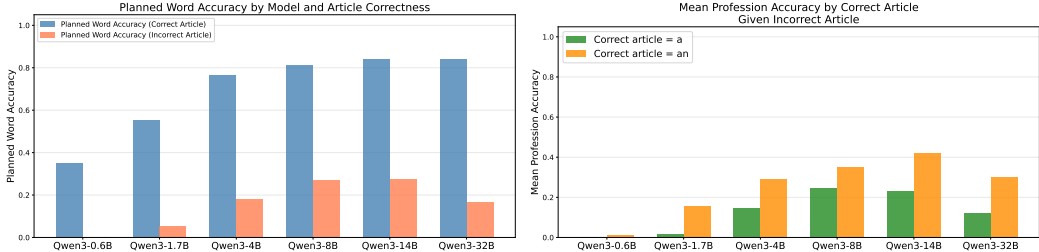

Figure 13: **Left**: Planned word accuracy, i.e. whether the model's predicted word matches the intended word, when given the correct or incorrect article. Models above 4B in size are highly accurate when given the correct article ($> 80\%$), and even smaller model achieve moderate accuracies. Given the wrong article, accuracies are lower, but still non-zero, indicating that models may have a strong planning goal that prevails even when the word is at odds with the article. **Right**: Planned word accuracy given the wrong article, by correct article (*a* or *an*). Though accuracy is low, models succeed on both *a* and *an* examples, indicating that successes are not driven by one class.

Performance differs by task. On the *a/an* task (Figure 13, left), models have generally high accuracy ($> 0.6$) when given the correct next token, but lower accuracy when given the incorrect one; the

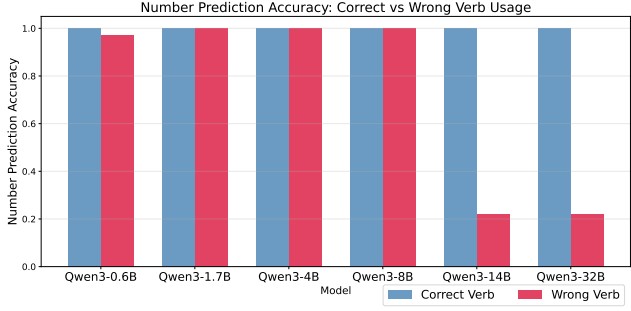

Figure 14: Number accuracy, i.e. whether the model's predicted number of animals matches the intended number, when given the correct or incorrect verb (*is / are*). Notably, small models produce the correct number regardless of whether they are given the correct or incorrect verb. In contrast, Qwen3-14B and 32B have starkly reduced accuracy when given the wrong verb form.

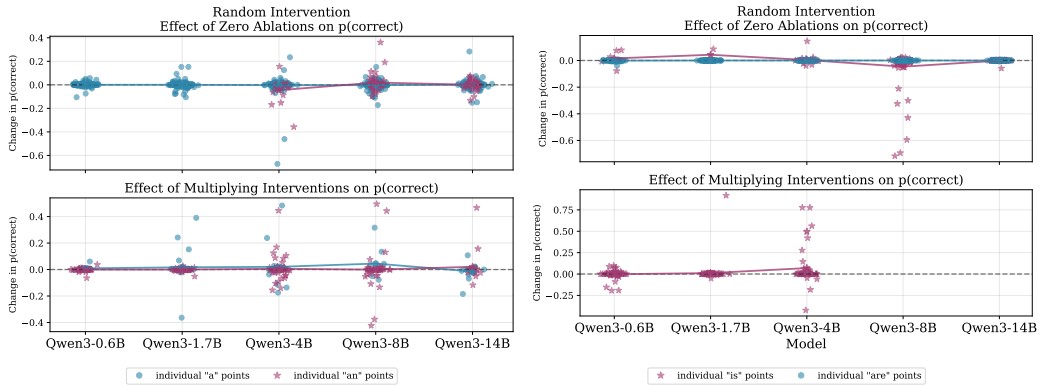

Figure 15: Effects of random features interventions on the *a/an* (**left**) and *is/are* (**right**) tasks. Neither intervention has a large effect on either dataset, indicating that our interventions do not succeed by random chance.

highest scoring models in that scenario achieve an accuracy of 0.3-0.4. Baseline accuracy here is in theory near 0, as models can predict any word. This suggests that although models are not always planning for the precise word we intend (and indeed, there are cases where we find no nodes corresponding to the planned word) they often are. And in some cases, they plan so strongly for the intended word that they output it even when it conflicts with the article.

This trend is much stronger on the *is/are* task. Our results from the analogous experiment (Figure 14) show perfect accuracy for all models when the correct verb form is given. Given the incorrect article, smaller models are (near-)perfectly accurate at predicting the correct number, but larger models (Qwen3-14B and 32B) perform much worse. This provides strong evidence that small models can perform the task (and that a lack of task abilities does not underlie their poor planning performance). However, the root of the behavior of large models is less clear. They appear to be more sensitive to (subject-verb) agreement, and thus produce outputs that agree with the number of the verb; in particular, given *is* as an incorrect next token, they tend to output *1*, rather than a number that agrees with the original animal quantities. In contrast, weak models do produce outputs like *. . . now there are 1 dog remaining.*

## F    RANDOM INTERVENTIONS

In order to ensure that our have not succeeded by random chance, we perform all-effects ablations on random active features in our *a/an* and *is/are* datasets. For an example where we normally intervene on $n$ features, we sample another $n$ features from the pool of all active last-position features, and record the effects of the intervention. The results (Figure 15) indicate that these random interventions are ineffective: neither the zero ablations nor the multiplying interventions work.

## G    ANIMAL PROBING AND INTERVENTION EXPERIMENTS

As done by Dong et al. (2025), we set up probing experiments as follows. We take 1000 stories from the validation set of Tinystories, and extract the first sentence. We then feed each first sentence to the model in the following prompt: `Here's the first sentence of a story: {sentence1}. Continue this story with one sentence that introduces a new animal character.` We then generate (greedy sampling) a next sentence, and recorded the animal contained therein.

We then filter this data down to only the datapoints containing the top-4 most common animals; typically, this leaves 600 or more examples. We then split the data 60/20/20 into train, validation, and test, and collected (transformer layer output) activations from the last token of each prompt. We then train a single-layer MLP probe to predict the animal that the model would predict, from these activations. We use a hidden dimension of 64 for our MLPs, as Dong et al. report that performance plateaus at $d = 64$. We run this analysis on all Qwen3 models, as well as on Llama-3-8B-Instruct,

used by Dong et al., and report results across hidden layers in. Figure 16 shows that our results on Llama-3 (8B) are similar to the original findings, with high F1 scores (0.6-0.7) across all layers but the first. Probing results for other models are varied; Qwen3-8B and 32B perform well (F1 near 0.6), while other models exhibit middling performance (F1 $< 0.5$).

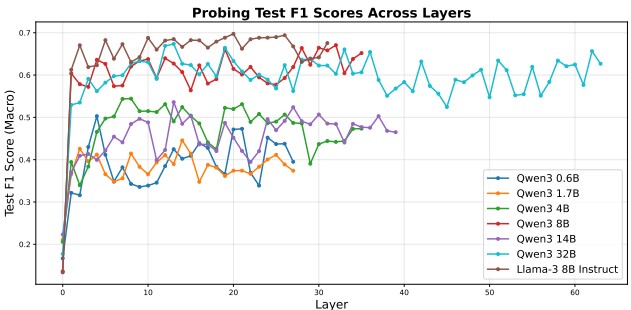

Figure 16: Macro F1 scores when probing models' last-token representations for the animal that the model will output in the following sentence, by model layer. F1 scores are high for certain models—Llama-3 (8B) and Qwen-3 (8B/32B)—but notably lower for others.

We then verify the causal relevance of the features found by these probes. If the probe has found a causally relevant feature at the end of the prompt that determines the animal that is output, altering that feature should alter the animal that is output. There are a variety of interventions that could be used to verify the features found by the probe: Ravfogel et al. (2021) intervening by reflecting representations across probe decision boundaries, while Giulianelli et al. (2018) compute the gradient of the probe's prediction (error) with respect to the model representations, and update the representations based on this. We could also use less probe-specific interventions like difference in means (Marks & Tegmark, 2024).

We opt for a simpler intervention: we pair each prompt in our dataset with a random prompt that led to the production of a distinct animal. We then generate a continuation to the first prompt, but patch the last-token activations of the second prompt onto the last token of the first prompt. We do so at all layers, effectively replacing all model activations at this position. This means that the next generated token is guaranteed to be the next token of the second prompt; furthermore, attention back to the patched position will receive the patched values. Since we have patched all possible layers in which the relevant features could reside, this intervention should cause the model to produce the animal from the second prompt, if the probed features are relevant. We perform this intervention across the same set of models as the previous experiment, with the exception of Qwen3-32B, as it is not supported by TransformerLens (Nanda & Bloom, 2022), the interpretability framework we use.

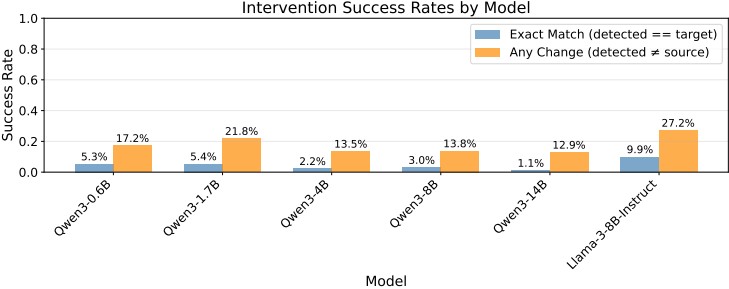

Figure 17: Success rates of our patching intervention, where we patch one prompt ($p_2$)'s last token activations onto another ($p_1$)'s last token during generation. We report both exact match (True if the output animal is $p_2$'s animal) and any change (True if the output animal differs from $p_1$'s original animal). In general, exact match is low, below 10%, while any change is higher, but under 30%.

Our results (Figure 17) suggest that the features found are not highly causally relevant. In relatively few cases ($< 10\%$ for all models) do we observe the output animal change that of the second prompt. In fact, in the majority of cases, the output animal does not change at all. This seems to be a violation of our Condition 1, that the found feature must have a causal impact on the model's planned token.

We note, however, that Llama-3 (8B), the only model from Dong et al. that we test, does have higher intervention efficacy. Moreover, if there are multiple features relevant for planning the animal to be produced, it would be necessary to find and intervene on all of these to produce a strong effect.

Despite this, we find it unlikely that planning takes place in this scenario. This is because the continuations corresponding to each animal output are generic: they do not hint to animal that will be produced. Consider, for example, the prompt and continuation *Mia and Dad were busy polishing their car. . . As they worked, a small, curious **fox** darted into the garage, tail wagging playfully.* The left context of *fox* imposes few constraints on the animal that is to follow; many animals can be *small* and *curious*. This hints that our Condition 2 may not be fulfilled here either: the model does not actually have to plan / prepare a context that licenses the animal eventually output.

# H  COUPLET CIRCUIT DETAILS

## H.1  COUPLET DATASET AND SAMPLES

Our couplets dataset was created by prompting Qwen-3 (32B) with the prompt "`/no_think You are a creative poet. Produce ONLY the first line of a rhyming couplet about the topic: '{topic}'. Return a single poetic line and nothing else.`". We sampled 5 couplets per prompt, kept only the first line of each couplet, and manually filtered these for well-formedness. We used the following topics: love (romantic, familial, self-love); death and mortality; coming of age; war and conflict; nature and the environment; home and belonging; identity and self-discovery; joy and happiness; anxiety and fear; loneliness and alienation; nostalgia and memory; hope and despair; anger and frustration; family bonds (siblings, parent-child); friendship; betrayal and trust; first love; marriage and commitment; loss of loved ones; childhood memories; aging and growing older; seasons and cycles; historical moments; the passage of time; immortality and legacy; justice and injustice; freedom and oppression; cultural identity; social alienation; tradition vs. modernity; community and belonging; weather and climate; animals and wildlife; urban vs. rural life; travel and journeys; food and taste; colors, sounds, and textures; dreams and aspirations; perfection and imperfection; truth and deception; power and powerlessness; spirituality and faith; art and creativity.

We then sampled couplet completions from the Qwen-3 models using the prompt: "`/no_think Write only the next line of this rhyming couplet: {first_line}`".

For convenience, we provide 5 example first lines of couplets and completions below.

---

**First line:** Two hearts entwined where vows take root,

**Qwen3-0.6B:** And where the stars are born, they sing.
**Qwen3-1.7B:** two souls forever bound in love.
**Qwen3-4B:** A love that grows beyond all doubt.
**Qwen3-8B:** A love that time cannot undo.
**Qwen3-14B:** A love that grows in tender fruit.
**Qwen3-32B:** A love that time and distance can't undo.

---

**First line:** Whispers of dread in the quietest breath,

**Qwen3-0.6B:** A shadowed path to the dark, where the night is kind.
**Qwen3-1.7B:** And shadows that dance in the darkest keep.
**Qwen3-4B:** Shadows creep where the light dare not reach.
**Qwen3-8B:** A shadowed path where no light dares tread.
**Qwen3-14B:** A shadow that creeps through the darkness of death.
**Qwen3-32B:** Shadows conspire as the night takes its death.

---

**First line:** Whispers of wind through the emerald trees,

**Qwen3-0.6B:** The moonlight dances on the water's edge.
**Qwen3-1.7B:** Soft as dreams, they drift and sigh.
**Qwen3-4B:** Singing the hush of a summer breeze.
**Qwen3-8B:** Sing a lullaby to the sleeping breeze.
**Qwen3-14B:** Softly they sing as the daylight decreases.
**Qwen3-32B:** Singing soft lullabies to the leaves.

---

**First line:** Whispers of winter kiss the thawing spring,

**Qwen3-0.6B:** The frost has passed, and the snow has gone,
**Qwen3-1.7B:** and silence bows to the bloom of the green.
**Qwen3-4B:** Blossoms rise in quiet, fragile song.
**Qwen3-8B:** Beneath the snow, the seeds begin to sing.
**Qwen3-14B:** Blossoms awaken where the frost once clung.
**Qwen3-32B:** Melting dreams in a tender, fleeting sting.

---

**First line:** A promise made in shadows, torn by dawn's deceitful light,

**Qwen3-0.6B:** A shadowed vow, once kept, now lost,
**Qwen3-1.7B:** Yet still, the heart remains true to what it
**Qwen3-4B:** A vow now lost in the glare of day.
**Qwen3-8B:** A vow once whispered fades into the night.
**Qwen3-14B:** A heart once whole now fractured, lost in endless, silent night.
**Qwen3-32B:** A truth once held so sacred, now lies shattered in the fight.

## H.2 CIRCUIT VERIFICATION

Here, we discuss the experiments performed to verify our circuit, which we claim acts via end-of-line, near-end-of-line, and rhyming features. We test these features as follows:

**End-of-line (EOL) features** We define EOL features as those where 7 out of the feature's top 10 activations immediately precede a token containing a newline, e.g. ".\n". We test that 1) activating these features prior to the end of the second line causes models to end the line prematurely, 2) deactivating these features after the model has completed the second line with a rhyme causes models to continue the line, instead, and 3) deactivating these features at the end of the first line causes the model to fail to rhyme, as EOL features regulate its attention to rhyming features.

For each of these experiments, we identify EOL features on each example as those that are active on the last word of the couplet's first line. We perform each experiment only on those couplets for which we have performed attribution. For experiment 1), we provide the model with the couplet's first line and the first 2 tokens of the second. We set the EOL features to 5 times their original values at the final position of this input, and any generated positions. We then allow the model to generate, using greedy sampling. We record the length of generation (in tokens) before the model outputs a newline, and compare it to the length of the original line.

For experiment 2), we provide the model with each entire, completed couplet (stripping any punctuation at the end), and set all EOL features to -5 times their original values. We do this at the final position of this input, and any generated positions. We use the model to generate, using greedy sampling, and record the length of the model' new generation, compared to that of the original.

For experiment 3), we provide the model with the first line of each couplet, and let it generate the next couplet (with greedy sampling), while setting all EOL features to -5 times their original values at the end of the first line. We then record rhyming accuracy.

Figure 18A shows the results of experiments 1) and 2). Upweighting EOL features indeed causes models to end the second line early, resulting in a large negative difference in line length. In contrast,

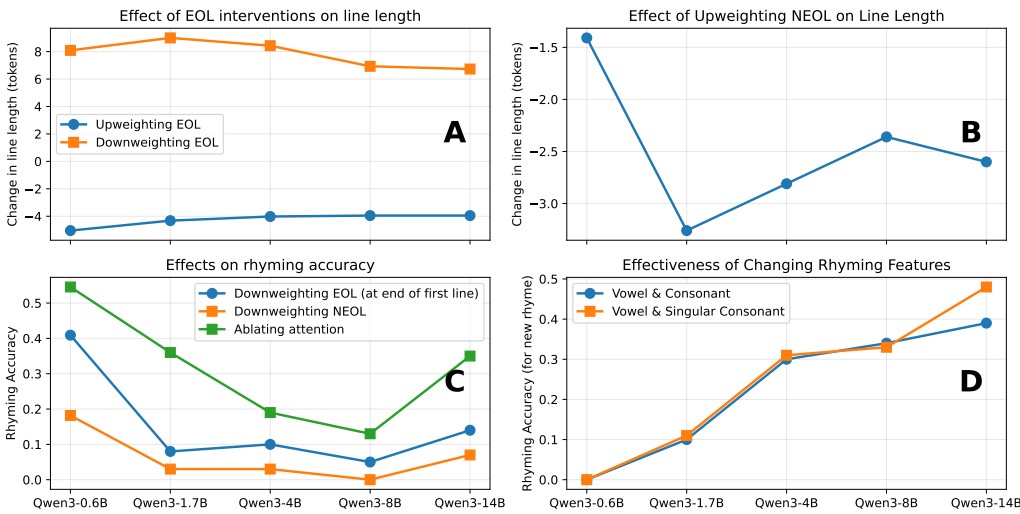

Figure 18: **A**: Effects of intervening on end-of-line (EOL) features. Upweighting them in the second line causes the line to end early, while downweighting them causes it to continue for longer than normal. **B**: Effect of upweighting near-end of line (NEOL) features in the second line. Upweighting these causes the model to emit a rhyme over 2 words earlier than normal. **C**: Effects of downweighting EOL features at the end of the first line, downweighting NEOL features in the second line, or ablating rhyming-relevant attention heads' patterns. The first two interventions drastically decrease the model's propensity to produce a rhyme, indicating that they help enable rhyming. The last is less effective, but still reduces accuracy far below the original, 100% accuracy. **D**: Rhyming accuracy when ablating original rhyming features, and upweighting those from another rhyme group. Larger models switch to the new rhyme group with 40% accuracy—lower than their original rhyming accuracy, but still relatively high.

downweighting said features prevents models from ever finishing a line, resulting in very long lines, relative to the original. Figure 18C shows the results of experiment 3). The rhyming accuracy for larger models (larger than 0.6B) is low, below 0.2; this is despite the fact that we perform the intervention on examples for which we have computed circuits, which are examples on which models succeed. This means that downweighting EOL features at the end of the first line seriously hindered performance. Moreover, qualitative inspection of the model's outputs showed no harm to the overall fluency of the completions, suggesting that this was not due to general harm to the model's abilities. This suggests that the EOL features do play an important role in regulating rhyming abilities, likely through the keys of attention heads, which tell them where to attend to.

**Near-end-of-line (NEOL) features** We define NEOL features as those where 7 out of the feature's top 10 activations occur 2-4 tokens before a token containing a newline, e.g. ".\n". We test that 1) activating these features at the beginning of the line causes models emit a rhyme early, and that 2) deactivating them stops models from rhyming.

For each of these experiments, we identify NEOL features on each example as those that are active on the second to last word of the couplet's second line, i.e. on the token before the rhyming word. We perform each experiment only on those couplets for which we have performed attribution. For experiment 1), we provide the model with the couplet's first line and the first 3 tokens of the second. We set the NEOL features to 5 times their original values at the final position of this input, and any generated positions. We then allow the model to generate, using greedy sampling. We record the length of generation (in tokens) before the model outputs a rhyming word, and compare it to the length of the original line.

For experiment 2), we provide the model with each couplet's first line, and set all NEOL features to -5 times their original values. We do this at the final position of this input, and any future positions. We sample a second line from the model using greedy sampling, and record rhyming accuracy.

Figure 18B shows the results of experiment 1). Upweighting NEOL features causes models to rhyme early - over 2 tokens early, for models above 0.6B. This suggests that the NEOL feature is causally responsible for models' output of a rhyming token. We note that this intervention qualitatively frequently caused models to rhyme not just early, but also rhyme often: models sometimes output multiple rhyming words (e.g. *Beneath the gray lay stray*), as if the need to rhyme (like the upweighting of the NEOL feature) was ongoing.

Figure 18C shows the results of experiment 2). The rhyming accuracy for larger models (larger than 0.6B) is low, below 0.2, just like when we downweighted EOL; indeed. Once more, this is despite the fact that we perform the intervention on examples for which we have created circuits, which are examples on which models succeed. Downweighting NEOL features in the second line thus seriously harmed performance. Similar to before, qualitative inspection of the model's outputs showed no harm to the overall fluency of the completions. Since these features act at the position where rhyming occurs, we hypothesize that they affect the queries of attention heads that would otherwise bring features over rhyming information, allowing models to then predict a rhyming word.

We thus also test this attention head theory. We record each model's attention during a normal forward pass, and when the NEOL features are strongly (-6x) downweighted. We then find the top-5 heads whose attention back to the end of the first line is reduced most by this ablation, averaged across couplets. We hypothesize that these heads play a causal role in rhyming abilities. Thus, we perform couplet generation as in the prior experiment, but transfer all of these 5 heads' attention back to the end of the first line, to the BOS token; we observe that this is what happens upon ablation, and such tokens are generally considered to be attention sinks. We then record rhyming accuracy.

Figure 18C shows the results of this experiment as well. This ablation is less effective than directly intervening on NEOL features directly; rhyming accuracies are 20-30% higher, though far below the 100% accuracy models achieved on the sentences for which we computed circuits. It is also significantly more targeted: we only alter 2 attention probabilities in 5 heads, rather than targeting many features.

**Rhyming features** As discussed in the main text, we find rhyming features using a heuristic. We look for features that activate on short tokens (all top-activating tokens are <5 characters) that are distinct (no more than 5 occurrences of the same token), and where 7 of the 10 top activating features

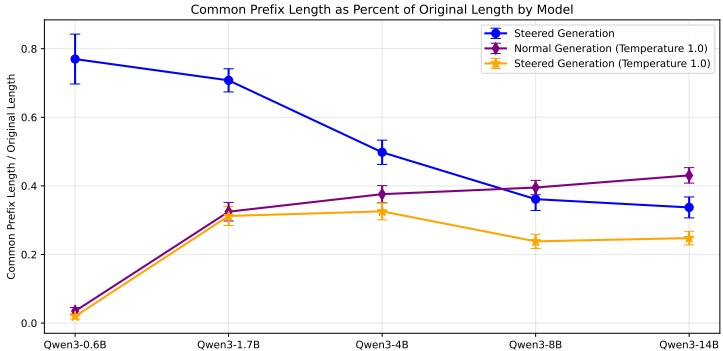

Figure 19: Length of the shared prefix between the original generation, generations with temperature 1.0, and steered generations, both greedily sampled and with temperature 1.0. Error bars show SE.

either start with the same vowel, or end with the same consonant. Manual inspection suggested that this yields relatively high-precision but only moderate-recall recovery of these features.

We test that these features control the output rhyme, by deactivating each example's original rhyming features at the end of the first line of the couplet, and upweighting the rhyming features of an example with a different rhyme. The resulting line should rhyme with the new example, not the original.

As in the main text, our results (Figure 18D) suggest that these features are indeed responsible for choosing the rhyme. Although accuracy is lower than on the original rhymes, it is moderate, and near that of the models overall on rhyming couplets.

### H.3 Intervening on Rhyming Features Changes Intermediate Tokens

We can test whether the intermediate context generated by the model changes at all upon rhyming feature intervention. To do this, we take the original generation of the model on a couplet, and its generation when its rhyming tokens are steered as in Section 5.3. We then record the length (in tokens) of the longest prefix shared between the original and steered generation. As baselines, we also compute the overlap between the original, greedy generation, and generations (both steered and unsteered) that we sample with temperature 1.0. If the rhyming features are genuinely causing the model to plan for future tokens, we should expect them to cause the model's intermediate tokens to change, more than temperature-based sampling would.

Our results (Figure 19) indicate that steering affects the intermediate context between the first line and the rhyming token output. For smaller models (0.6B and 1.7B), this intervention do no more than simple sampling does. But for larger models (8B and 14B), the effect of this intervention upon generations exceeds that of normal sampling—even when combining the intervention with greedy decoding. Thus, the intervention alters both the intermediate and final tokens that models output.

## I Potential Local Planning Features

To find local planning features in couplet circuits, we search for features in our circuit that upweight the rhyming word that is eventually output, or have one of their top-10 activations on that word. We search at the position before that word is output; that is, we look for *say X* features that are causally relevant even before the model outputs *X*. For a circuit in which this could be occurring, see Figure 20.

For each model, and each of its top-10 words by number of *say X* features, we steer on those *say X* features, setting their activations to 3, 5, or 7 times their original values. We do so on 5-15 token fragments of sentences from the TinyStories dataset (Eldan & Li, 2023)—a neutral context where models are not likely strongly planning. We then record whether each model eventually output *X*, and qualitatively inspect the outputs.

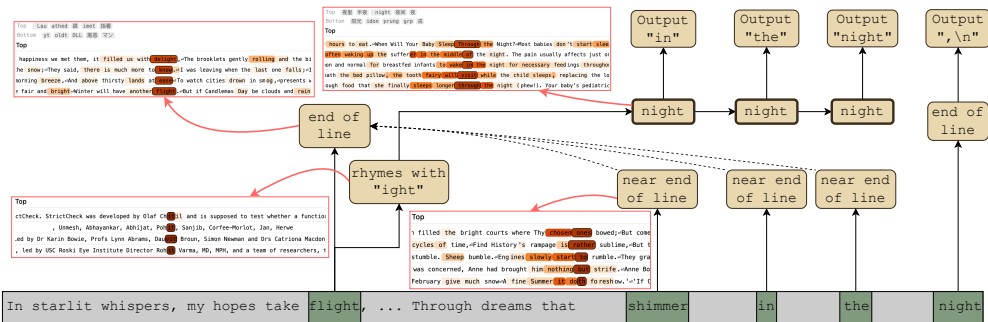

Figure 20: A feature circuit for a couplet ending in *in the night*. Unlike prior circuits, this circuit involves a specific *night* feature that drives the production of the phrase *in the night*.

Qualitatively, we observe that steering can lead to the sorts of sensible generations we observed in the poetry setting. Steering on the *say "night"* feature leads to generations such as *One day, a girl named Mia went for a walk.* **She saw a cat and started to follow it** to turn into *One day, a girl named Mia went for a walk.* **She saw a cat in the night**; notably, *in the* appears to specifically license *night*. Similarly, steering on the *say "dream"* feature often leads to outputs like *recurring dream* or *American dream*, i.e., contexts that are specific to *dream*.

However, this is not always the case. Steering too hard can cause the model to output the target word even in infelicitous contexts, or to only output the word; some *say X* features seldom produce the target word when steering. How can we measure whether the steering not only (1) produced the word *X*, but also (2) maintained a coherent sentence (up to the point where the word *X* was output) and (3) truly adapted the context to license *X*? We can measure (1) programmatically, but (2) and (3) are harder. For coherence, we query Claude Sonnet 4.1 about the coherence of each steered generation (Listing 1); to verify that Claude is a good judge of coherence we annotate 100 examples for coherence, and find high agreement (80%, where most disagreements come from Claude missing incoherence).

To estimate models' abilities at (3), we filter examples to include only those where (1) and (2) are fulfilled; we also filter out any examples where the original and steered generation are identical up until the word *X* is output, as adaptation has surely not occurred in such cases. Then, we estimate how many of these examples fulfill (3) by manually annotating 100 examples per model for whether they contain context adaptation that could indicate planning. For example, we look for phrases like *her own* when steered towards *own*, or *had a recurring dream* when steered towards *dream*, when the original generation did not contain similar phrases. We also mark as incorrect examples that are ungrammatical / incoherent, but were missed by Claude in the first round of filtering, as a model that is successfully adapting its context for a planned token should not produce such outputs. We then plot these metrics.

Our results (Figure 21) indicate that larger models are more successful at steering towards *X* and more likely to adapt their context to match *X*, though they are no more coherent than smaller models. However, few examples actually fulfill all of these conditions: even in Qwen-3 (14B), only 10% of examples do so. So, while we believe that these features may be part of a generalized phenomenon whereby models plan for words by boosting $n$-grams that end in those words, our uncertainty is rather high. Our current hypotheses still rely on qualitative evidence, and more study is needed to understand the precise mechanisms by which these features work, and more consistently elicit and measure planning behavior from them.

Listing 1: Claude 4.1 Sonnet Prompt. Note that we only employ the coherence judgments, as the contains-word criterion can be checked programmatically, and we found Claude's adapts-context responses unreliable.

```
f"""I need you to analyze a text generation where a model was steered
    to include a specific word.

<input_prompt>
{input_text}
```

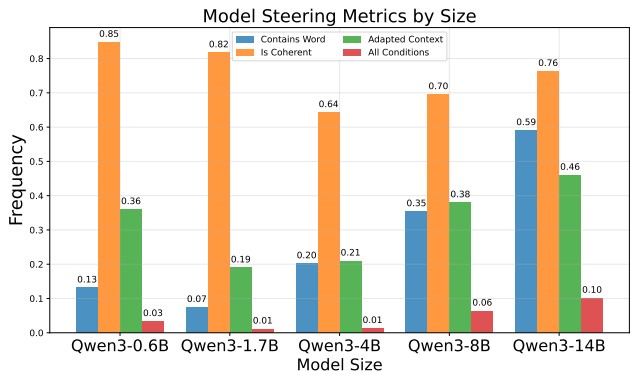

Figure 21: Steering metrics by model size, averaged across steering strength. Overall, as model size increases, the ability to elicit *say X* by steering increases. The model's tendency to adapt their context also appears to increase with size. Coherence is mixed, and appears unrelated to model size. Ultimately, few examples fulfill all three conditions—10% in the best case.

```
</input_prompt>

<baseline_generation>
{baseline_generation}
</baseline_generation>

<steered_generation>
{generation}
</steered_generation>

<steered_word>
{steered_word}
</steered_word>

Please analyze this generation and answer the following questions:

1. **Contains steered word**: Does the steered generation contain the
     exact word "{steered_word}"? (Look for exact match, case-
     insensitive)

2. **Coherence**: Is the steered generation coherent up to the target
     word. Is it natural, or does it show signs of breakdown such as:
   - Excessive repetition of words or phrases
   - Unnatural/nonsensical sentences
   - Abrupt topic changes that don't make sense
   - Grammatical breakdown
   - IMPORTANT: Focus on coherence up to the point where the steered
       word appears (or would appear). Ignore any incoherence that
       happens after the steered word.

3. **Context adaptation**: Compare the steered generation to the
     baseline generation, starting from the input_prompt. Did the model
      modify the words immediately preceding where the steered word
     appears (or would appear) compared to the baseline? Look for
     changes like:
   - Adding/changing articles ("a", "the", "an")
   - Adding/changing prepositions ("in", "on", "at", "with")
   - Changing verb forms or tenses
   - Adding adjectives or modifiers
   - Any other modifications that help the steered word fit better

IMPORTANT: For context adaptation, you must compare the steered
     generation against the baseline to see if modifications were made
     specifically to accommodate the steered word. Count the number of
```

```
          words that were added or changed immediately before where the
          steered word appears (or would appear).

      When counting adaptation words:
      - Count new articles, adjectives, adverbs, or prepositions added
          before the steered word
      - Count changed words that help the steered word fit better
      - Do NOT count the steered word itself
      - Focus on the immediate context (usually 1-3 words before the steered
          word)
      - If the steered word was not present, there is no adaptation

      Example 1:
      input_text: "Once upon a time, there was a"
      baseline: "Once upon a time, there was a young girl named Lily who
          lived in a small village"
      steered_word: "day"
      steered_generation: "Once upon a time, there was a day when all the
          day of day day day day"
      contains steered word: yes
      coherence: yes (the generation is coherent up to "day")
      contain adaptation: no (day is inserted instead of young)

      Example 2:
      input_text: "Once upon a time, there was a little boy named"
      baseline: "Once upon a time, there was a little boy named Timmy who
          loved to play with numbers. One"
      steered_word: "dream"
      steered_generation: ""Once upon a time, there was a little boy named D
          . He had a recurring dream that he had""
      contains steered word: yes
      coherence: yes (the generation is coherent up to "day")
      contain adaptation: yes (sets up dream with "he had a recurring")

      Please provide your analysis in this exact format:

      <analysis>
      <coherence_explanation>Brief explanation of any coherence issues found
          </coherence_explanation>
      <adaptation_explanation>Brief explanation of any context adaptations
          made compared to the baseline</adaptation_explanation>
      <contains_word>yes/no</contains_word>
      <is_coherent>yes/no</is_coherent>
      <adapted_context>yes/no</adapted_context>
      <adaptation_word_count>number (0 if no adaptation, otherwise count of
          adapted words)</adaptation_word_count>
      </analysis>"""
```

## J COMPARISON OF INSTRUCTION-TUNED AND BASE QWEN-3 MODELS ON *A-An* AND *Is-Are* TASKS

Throughout this paper, we have analyzed the behavior of instruction-tuned Qwen-3 models. However, it is unclear how the performance and mechanisms of said instruction-tuned models differs from that of base models. We investigate this by running the Qwen-3 Base models, from 0.6B to 14B, on the *a/an* and *is/are*. For the latter task, we reformulate the base model's input to exclude the instructions present in the original task (*Repeat this sentence and complete it.*), though we find this to have little effect on the results.

Our results indicate that the base models outperform the instruction-tuned models on the *a/an* task slightly (Figure 22, left): recall of the majority-article *a* is similar between the two, while the base models have consistently higher recall of *an* at 1.7B parameters and above. By contrast, on the

*is/are* task (Figure 22, right), the instruct models significantly outperform the base models: while both achieve high recall on *are*, only the instruct models eventually achieve high recall on *is*. This is despite the fact that base models, too, are numerate enough to complete the task: even when given the wrong verb, they output the correct number, much as the instruct models did.

These results indicate that neither model variant—base or instruction-tuned—strictly outperforms the other. Rather, differences seem driven more by the data distribution: the *a/an* task is more akin to pre-training data, while the *is/are* task involves math, a focus of instruction-tuning.

## K PLANNING FOR MEASURE WORDS IN CHINESE

The most successful examples of backward planning are English agreement tasks, which raises the question: can models perform backward planning in other languages or contexts? The relative weak performance of Qwen-3 (14B) and smaller models makes adding complex tasks challenging, but we can test agreement abilities in Chinese, in which Qwen models perform well.

We focus on the phenomenon of measure-word agreement in Chinese. Chinese uses measure words, which function analogously to e.g. the word *loaf (of bread)* in English: rather than saying *one bread*, one says one *loaf of bread*. Similarly, in Chinese, *one person* becomes 一个人 (one [person-unit] person), while *three pigs* becomes 三头猪 (three [livestock-unit] pigs). Notably, different nouns require different measure words, but the measure word precedes the noun, much like *a/an*.

We can thus ask: given a context that indicates that a given noun will appear, do models engage in forward planning for that noun? And do they also engage in backward planning to determine which counter word should be used? Owing to our own limited knowledge of Chinese, we construct a smaller set of 10 examples, which elicit distinct measure words:

1. 他看见四位骑士骑着四…匹马。: He saw four knights upon 4…[measure-word] horses.
2. 这套公寓有三…间卧室。: The apartment had 3…[measure-word] bedrooms.
3. 他看见四位农夫赶着四…头牛。 He saw four farmers with four…[measure-word] cows.
4. 他看见四位调酒师端着四…杯鸡尾。: He saw four bartenders carrying four…[measure-word] cocktails.
5. 她听到笼子里传来歌声。进去一看，里面是一…只鸟。: She heard singing from in the cage. Inside she saw a…[measure-word] bird.
6. 剧团刚刚表演完一…出话剧。: The theater troupe had performed a…[measure-word] play.
7. 大森林里长着1000…棵树。: In the large forest, there grew 1000…[measure-word] trees.
8. 池塘里游着四…条鱼。: In the pond, there swam four …[measure-word] fish.
9. 这位艺术家创作了十二…幅画。: The artist created twelve…[measure-word] paintings.
10. 夜空中，他们看到了1000…颗星星。: In the night sky, they saw 1000…[measure-word] stars.

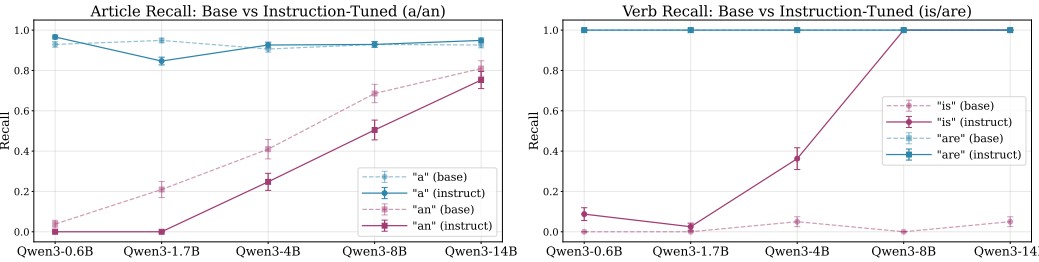

Figure 22: Performance of base (dashed line) and instruction-tuned (solid line) models on the a/an (left) and is/are (right) tasks.

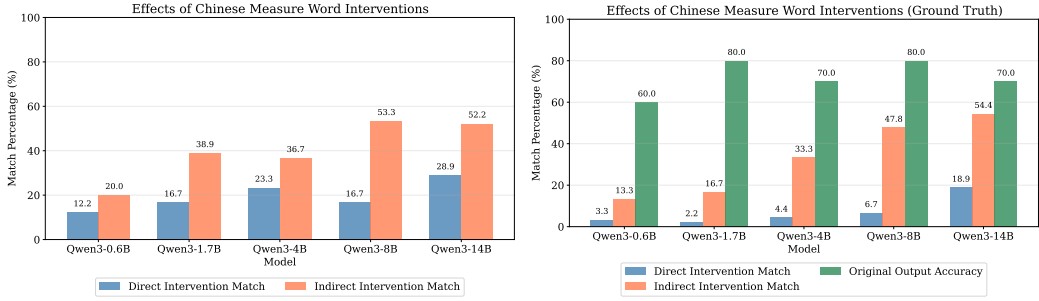

Figure 23: The effectiveness of replacement interventions, measured by the percent of cases where the intervention induced to model to output the expected the measure word. **Left**: the expected measure word is considered to be the measure word that the model output on the target example. **Right**: the expected measure word is considered to be the ground truth measure word from the list above—even if the model failed to output it originally. Models' accuracies with respect to the original ground truth are shown in green.

We select these examples because preliminary testing indicated that at least larger models produce the expected counter words when given them; they do not all have a correct answer per se. Thus, unlike in the *a/an* and *is/are* cases, we cannot treat these as a behavioral evaluation of backward planning ability. However, we can still analyze models' circuits.

Thus, we compute attribution graphs for these examples, attributing back from the predicted (measure) word. As in the *a/an* example, we find features that correspond to our intended word. For example, planning features that fire on and upweight *bird* activate on example 5, while *painting* features activate on example 9. However, analogues to *say a/an* features are often absent from these circuits: while "say [any measure word]" features are common, "say [specific measure word]" features are not. That is, while there are "say head [of cattle]" features that correspond to the livestock measure word 头, not all measure words have these. How do these features contribute to model outputs?

We test this by running interventions, as in the couplets section of the main text. Specifically, we run the model on a given *target* example, but downweight its planning features, while upweighting features from another *source* example; we run these replacements for all source × target combinations. We aim to change the model's output measure word to the measure word of the target example. In addition to performing unconstrained interventions, we also perform direct-effects interventions. These test for the presence of "say [specific measure word]" features that might be hidden in transcoder errors: if direct-effect interventions fail, while full-effects experiments succeed, we know that intervening features exist in downstream MLPs, even if the transcoders miss them.

Our results indicate that this intervention is relatively successful. Figure 23 (left) shows the proportion of examples in which performing the intervention successfully changed the word into the expected measure word, i.e. the word that the model originally output on the target example. Performance generally increases with scale, increasing from 20% match with the expected measure word to 52.2%. By contrast, direct-effects interventions produce effects around half the size. When we use the ground-truth measure word as our expected measure word (Figure 23, right), smaller models perform worse, direct effects interventions grow less effective and the scaling trend becomes more obvious.

We conclude that the planning nodes do also control the measure word and noun output. However, we note that the direct-effects interventions have moderate effects, varying by measure word. We hypothesize that this is because some Chinese measure words have semantics that closely align with their corresponding noun; for such examples, the planning features, along with "say [any measure word]" features suffices to upweight the measure word.

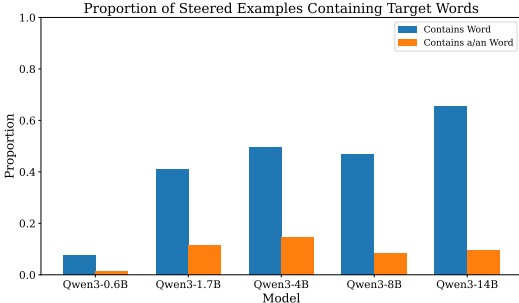

Figure 24: Results of steering on planning features from the *a/an* task. For each model, we plot the proportion of examples where the model produces the planned word (blue) and the proportion where the model produces *a* or *an*, followed by the planned word. Models often produce the planned word when steered, but less often produce the article in front of it.

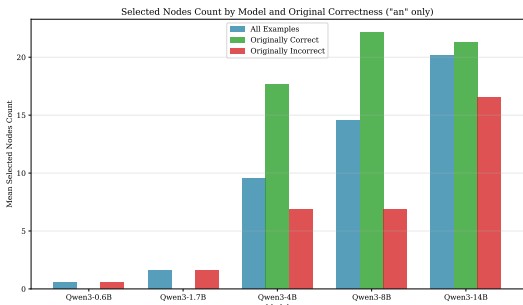

Figure 25: Planning feature counts on *an* examples, by model and correctness. The smallest models have few planning features overall, while the largest has relatively many in both the correct and incorrect cases. In the 4B and 8B parameter models (with nascent circuits), there is a large gap between the number of planning features active in correct and incorrect cases..

## L  STEERING ON *A-An* PLANNING FEATURES

In this section, we provide additional evidence that planning features, such as those in the *a/an* experiments, do not directly cause the upweighting of said indefinite articles. To do so, we take the same approach in Section 5.4 to identify planning features to steer on. We then use them to steer models on the TinyStories dataset. We then check for the presence of the planned word, and for the presence of *a/an* before it. Our results (Figure 24) indicate that while steering is effective at eliciting the desired word, this does not entail producing *a/an*.

## M  WHAT ARE NASCENT CIRCUITS?

We find that Qwen-3 4B and 8B have nascent circuits for *a/an* and *is/are*, leading them to achieve middling recall of the minority classes of those tasks. But what does it mean for circuits to be nascent, or not fully formed? There are a few points of potential breakage in the circuit:

- **Planning Features** The models could lack planning features, or fail to activate them sufficiently.
- **Downstream Connections** The models could lack connections from planning features to downstream features upweighting *a/an* or *is/are*

We investigate the first hypothesis in the *a/an* task, by counting the number of planning features active on *an* examples, distinguishing cases where model outputs were correct and incorrect. Plotting these (Figure 25) shows markedly different behaviors across model scale. The 0.6B and 1.7B parameter models have very few active planning features overall. The 4B and 8B parameter models have many active planning features in the correct case, but notably fewer in the incorrect case.

Finally, Qwen-3 (14B) has many planning features active in both cases, though slightly fewer are active in incorrect cases.

This suggests that the failure of these nascent circuits is due at least in part to the models' failure to adequately activate planning features.

