# OpenReview forum: "Latent Planning Emerges with Scale"
_ICLR.cc/2026/Conference — ICLR 2026 Poster_

### Official Review · Reviewer_JcTR · 2025-10-31

**Soundness:** 4
**Presentation:** 4
**Contribution:** 4
**Rating:** 8
**Confidence:** 3

**Summary:**

Investigates whether LLMs engage in latent (not explicitly generated) planning and shows that planning ability grows with model scale (in the Qwen-3 family).

Contributions

* Provides a causal definition of latent planning  -- planning is an internal representation that (i) causes the model to produce a specific future token for forward planning and (ii) causes generation of a context that makes that token more likely. This improves on purely observational/probing definitions.
* Provides simple agreement tasks as planning probes: On a/an, is/are, and el/la tasks, larger Qwen-3 models reliably plan ahead for the content word and use that to choose the right function word. Smaller models show nascent but incomplete mechanisms.
* Mechanistic evidence via transcoder feature circuits that identify “planning features” that represent the future word and show, through interventions, that ablating them hurts performance and boosting them helps, indicating genuine causal relevance.
* Causal-mech interpretability recipe for monitoring such emergence of forward planning and backward planning ability in  open models.

**Strengths:**

* Originality: Introduces a causal definition of latent planning that distinguishes between forward (goal-directed token production) and backward (context-shaping) planning. This causal approach is  novel in how it rephrames what “planning” means for decoder models and correcting an overextension in prior work that equated decodability with intent. The integration of transcoder feature circuits with causal interventions is also a novel methodological synthesis, enabling verifiable mechanistic evidence rather than speculative probing.
* Quality: The experiments are rigorous and well controlled. The progression from simple grammatical-agreement tasks to rhyming and prose-steering scenarios is also structured well logically and empirically. The use of quantitative flow analysis within feature circuits adds an added layer of interpretability
* Clarity: Definitions are explicit, figures are clear and interpretable. The argument flows naturally from conceptual motivation to empirical validation.
* Significance: The results establish that latent planning mechanisms emerge with scale and that forward planning precedes backward planning—an interpretable scaling law that contributes to our understanding of model cognition.

**Weaknesses:**

* In terms of planning, the paper over-indexes on short-range linguistic dependencies. Not clear if this scales to true multi-step reasoning or action planning.
* Limited to Qwen-3 series, which hurts generalization
* Experiments provide only limited support for backward planning, and the analysis of whether the generated context “licenses” the planned token is overly qualitative. :

**Questions:**

* Can you add a quantitative measure of contextual dependency?
*  Do you believe these results generalize to multi-step cognitive planning or compositional reasoning? Could this be shown?

---

> ### Author Response · Authors · 2025-11-22
> **Author Response**
>
> Thank you for your positive review!. Below, we address some of your questions and concerns; see the rebuttal to all reviewers for more comments.
>
> > In terms of planning, the paper over-indexes on short-range linguistic dependencies. Not clear if this scales to true multi-step reasoning or action planning… Do you believe these results generalize to multi-step cognitive planning or compositional reasoning? Could this be shown?
>
> This is true! However, this is due to the fact that many open-source models perform poorly / inconsistently on complex planning tasks, hindering our attempts to interpret them. For example, chain-of-thought (CoT) unfaithfulness would have been one good real-life scenario to study, but we could not reliably elicit this from Qwen-3 models. As open source models become more competent, and as transcoders are trained on them, we will be able to study more realistic tasks.
>
> We think that our methods would scale to more realistic tasks - see the response to all reviewers for more details. As far as our results go, we are less certain. It seems likely that these also involve a planning feature that regulates downstream model behavior, driving it towards the planned output: Lindsey et al. (2025) observe this on a complex CoT unfaithfulness task, where the model plans to reason towards a given answer. However, we cannot currently test this with our methods, given the lack of transcoders for strong models. This might be possible using other methods - probing could scale to the largest open-source models, and could still enable us to find likely planning representations, and intervene with respect to them, at the cost of a lower-resolution picture.
>
> We want to point out that latent multi-step reasoning is still challenging for current models [1]. On the other hand,  as discussed in our response to all authors, investigating explicit multi-step reasoning (in models capable of it) should be possible with our methods. We would be excited to see work that attributes from directions in activation space representing higher-level quantities than token logits, e.g. a direction for “Why did my model produce this reasoning step?”
>
> [1]: Do Large Language Models Perform Latent Multi-Hop Reasoning without Exploiting Shortcuts? (Yang et al., ACL Findings 2025)
>
> > Limited to Qwen-3 series, which hurts generalization
>
> This is true, and a trade-off that we make in studying one model family across scales at using fine-grained feature circuits: there are no transcoders for other entire model families, and also no transcoders for individual models that are large / competent either. As transcoders are expensive to train, training a new suite of transcoders is not feasible. We could avoid this problem by using e.g. probing instead; this would work on almost any models, but we would lose the benefits of circuits and qualitative interpretability of transcoder features.
>
> Despite this, past work suggests that our findings may nonetheless generalize across models. Hanna et al. (2025) replicate many of Lindsey et al.’s (2025) findings, showing that many circuits from Claude also appear in Gemma-2 (2b). These range from multi-hop and multilingual reasoning, to analogies; they also find parallels between grammatical circuits in Llama 3.2 (1B). So, we believe that feature circuits may often be similar across models.
>
> > Experiments provide only limited support for backward planning, and the analysis of whether the generated context “licenses” the planned token is overly qualitative… Can you add a quantitative measure of contextual dependency?
>
> First, we want to highlight that in the couplet experiments, we do quantify the degree to which the generated context licenses the planned token - we measure both the degree to which steering causes model generations to diverge early from unsteered generation (Figure 19), and the degree to which steered context facilitates the production of a rhyme (Figure 5, right). However, our final experiment (Section 5.4) does rely on manual annotation of the model’s generation to determine whether it contains a context that licenses the planned word. Having given this some thought, the two metrics from before transfer poorly to Section 5.4.
>
> First, we measured the divergence of steered generations from the unsteered generation, and did find that they diverged early. However, not all divergence is good: if the model simply generates the steered word with no licensing context, this is also divergence. Measuring facilitation effects is tricky as well, as there is no clear baseline with which to compare p(steered-for word | steered context). When we compared this to p(steered-for word | input text), we almost always observed facilitation effects, but this comparison is likely too lenient.
>
> This said, we agree that finding a more quantitative way of measuring this specific instance of licensing, as well as of measuring licensing more generally across tasks and scenarios, is an important question!

---

### Official Review · Reviewer_nfGc · 2025-11-01

**Soundness:** 3
**Presentation:** 3
**Contribution:** 2
**Rating:** 4
**Confidence:** 3

**Summary:**

This paper examines whether large language models (LLMs) perform latent planning, that is, internally representing and reasoning toward future tokens without explicit plans. Using Qwen-3 models (0.6B–14B), the authors find that forward planning emerges with scale while backward planning remains limited, offering a causal framework and mechanistic evidence via transcoder feature circuits.

**Strengths:**

- The paper provides a clear and causally grounded definition of latent planning that distinguishes genuine planning from mere correlational predictability.

- The study offers comprehensive scaling insights, showing how planning abilities gradually emerge and strengthen as model size increases.

- It links mechanistic interpretability to AI safety, highlighting how latent planning could relate to hidden goal pursuit or “scheming”, thereby extending the work’s broader relevance.

**Weaknesses:**

- The chosen tasks, such as a/an prediction and rhyming couplets, are synthetic and narrowly scoped, limiting the conclusions’ applicability to real-world reasoning or planning.

- The evidence for backward planning is weak and inconclusive, raising doubts about whether full planning mechanisms have truly been demonstrated.

- The study lacks cross-model comparison, as it focuses only on the Qwen-3 family, making it unclear whether similar phenomena occur in other model architectures.

- Some of the causal claims may be overstated, since interventions could affect correlated linguistic or contextual features rather than genuine planning representations.

**Questions:**

- Could the observed causal effects arise from correlated features instead of true planning representations?

- How would the proposed framework generalize to complex goal-directed or multi-step reasoning tasks?

- What is the relative contribution of instruction-tuning versus model scale in the emergence of latent planning?

- How might this causal framework be applied in AI safety monitoring to detect latent scheming or hidden goal formation?

---

> ### Author Response · Authors · 2025-11-22
> **Author Response [1/2]**
>
> Thank you for your careful review! Below, we address some of your questions and concerns; see the rebuttal to all reviewers for more comments.
>
> > The chosen tasks, such as a/an prediction and rhyming couplets, are synthetic and narrowly scoped, limiting the conclusions’ applicability to real-world reasoning or planning.
>
> This is true! However, this is due to the fact that many open-source models perform poorly / inconsistently on complex planning tasks, hindering our attempts to interpret them. For example, chain-of-thought-unfaithfulness would have been one good real-life scenario to study, but we could not reliably elicit this from Qwen-3 models. As open source models become more competent, and as transcoders are trained on them, we will be able to study more realistic tasks.
>
> > The evidence for backward planning is weak and inconclusive, raising doubts about whether full planning mechanisms have truly been demonstrated.
>
> We contend that while backward planning is weak / inconclusive on the couplet task, our evidence on the a/an and is/are tasks is stronger - we can consistently see backward planning. However, we also emphasize that we do not claim that models possess full or correct planning mechanisms. Whether a given model plans on a given task is regulated by the model’s capacity, as well as the task’s complexity and its frequency / importance (in terms of training loss). Thus, larger models plan more, and common planning tasks are learned faster. So, if models’ planning abilities appear piecemeal or incomplete, this is not unexpected.
>
> > The study lacks cross-model comparison, as it focuses only on the Qwen-3 family, making it unclear whether similar phenomena occur in other model architectures.
>
> This is true, and a trade-off that we make in studying one model family across scales at using fine-grained feature circuits: there are no transcoders for other entire model families, and also no transcoders for individual models that are large / competent. As transcoders are expensive to train, training a new suite of transcoders is not feasible. We could avoid this problem by using e.g. probing instead; this would work on almost any models, but we would lose the benefits of circuits and qualitative interpretability of transcoder features.
>
> Still, past work suggests that our findings may nonetheless generalize across models. Hanna et al. (2025) replicate many of Lindsey et al.’s (2025) findings, showing that many circuits from Claude also appear in Gemma-2 (2b). These range from multi-hop and multilingual reasoning, to analogies; they also find parallels between grammatical circuits in Llama 3.2 (1B). So, we believe that feature circuits may often be similar across models.
>
>  > Some of the causal claims may be overstated, since interventions could affect correlated linguistic or contextual features rather than genuine planning representations… Could the observed causal effects arise from correlated features instead of true planning representations?
>
> It depends on what you mean by *correlated features*. It is not the case that the planning features cause the model to output e.g. a/an on their own. When we intervene on these features, but measure only their direct effects (that is, we do not let them affect downstream features) the effects are much smaller, indicating that their causal effect on the downstream “a/an” features are an important part of the models’ planning mechanism (Figure 3, right).
>
> Moreover, said planning features do not always cause *say a/an* features to activate. Just steering on these features in a non a/an context will often produce the planned word itself; whether or not a/an also appears depends on the context. For example, if we give Qwen-3 (1.7B) the input One day, a little girl named Lily found a needle, and steer on a house feature, the continuation is not simply a house, but in her house, a context-appropriate continuation that does not include an article at all. We have added an experiment demonstrating this in Appendix L.
>
> It may be true that the mechanism is a simple combination of two rules: “detect if a/an should be output” and “detect the word that should be output later”; if both rules are satisfied, then the appropriate “say a/an” features are activated. If this is what you mean, we agree that this simple mechanism is at work. However, note that we don’t claim that models have *true planning representations*, or do planning “correctly”. Rather, we aim to define what planning is, and how LLMs do it - even if their planning methods are not as robust or generalizable as we would like.

---

> > ### Author Response · Authors · 2025-11-22
> > **Author Response [2/2]**
> >
> > > How would the proposed framework generalize to complex goal-directed or multi-step reasoning tasks?
> >
> > We believe that our framework will be able to explain such phenomena, once open source models become competent enough to exhibit them. Sometimes, as in CoT unfaithfulness examples where the model plans for a single-token answer, it may suffice to simply attribute from the given answer token, as done here; one could then see the intermediate reasoning that led to that token.
> >
> > In other cases, e.g. detecting a hidden goal that drives model behavior without producing one “smoking gun” answer token, we may want to attribute back from more general high-level model actions, such as “Why did the model produce a refusal?” or “Why did the model make a given suggestion?”. Though past work has not yet done so, attributing from such higher-level actions seems possible; one must identify a direction in activation space corresponding to such an action, and attribute from this. We think that techniques like probing for such higher level actions, or using a difference-in-means-esque approach to identify causally relevant directions for such actions.
> >
> > > What is the relative contribution of instruction-tuning versus model scale in the emergence of latent planning?
> >
> > This is an interesting question, which we answer behaviorally by testing the Qwen-3 Base models on the a/an and is/are tasks. Qwen-3 Base slightly outperforms its IT’d counterpart on a/an; however, it performs poorly on is/are at all sizes, even when we remove the instructions from the inputs, making them more suitable for base models. This is despite the fact that Qwen-3 Base can perform math (predicting the correct numeral, given the wrong verb) at all sizes. As neither variant consistently outperforms the other, we attribute this more to the data distribution than to intrinsic differences between the two model variants: the a/an task looks like pre-training data, while the is/are task requires math abilities learned during instruction-tuning. We have added this to Appendix J.
> >
> > The differences between the base and instruction-tuned circuits are more difficult for us to measure mechanistically, as we only have transcoders for the instruction-tuned Qwen-3 models. While transcoders trained on base models generally transfer to instruction-tuned models due to similar activations, the Qwen-3 models we study (both base and fine-tuned) are independently distilled from larger base and instruction-tuned models. As a result, the base and instruction-tuned variants do not have the similar activations that would result from directly instruction-tuning the base models; this precludes transcoder transfer.
> >
> > > How might this causal framework be applied in AI safety monitoring to detect latent scheming or hidden goal formation?
> >
> > In the context of real-time monitoring, a naive approach would be to perform attribution on each output token of a given model; attribution gives us graphs with a causal interpretation, so causal interventions are likely not necessary for monitoring purposes. However, this naive approach would be infeasibly compute-intensive, as attribution is expensive. So, instead of attributing from every output token, we could train a probe to detect potentially suspicious / planning-oriented outputs, and only perform attribution on those; such probing classifiers are common in current safety approaches.
> >
> > Then, we need to determine if the circuit represents scheming / hidden goals. This would be easiest if we already had a set of known problematic hidden goal features; otherwise, the fact that planning features are not distinct from other non-planning features makes this difficult. This issue could be solved by the use of interpretability agents on feature circuits; an LLM agent could assemble the attribution graph into a circuit, and on the basis thereof, determine if the circuit indicates a hidden goal. We note that while we have seen preliminary testing on parts of this pipeline (finding circuits from attribution graphs), interpretability agents are still immature, so there is still work to be done to make this pipeline work in practice.

---

> ### Author Response · Authors · 2025-11-28
>
> Hello Reviewer nfGc - thanks for taking the time to review our paper! If you have a bit more time, we'd love to hear what you thought of our response, especially our correlated features experiment and the way our planning framework would work in more complex scenarios.

---

### Official Review · Reviewer_shSX · 2025-11-03

**Soundness:** 3
**Presentation:** 3
**Contribution:** 4
**Rating:** 4
**Confidence:** 3

**Summary:**

The paper's central hypothesis is that latent planning is an emergent capability that increases with model scale. It seeks to answer (1) whether LLMs engage in a mechanistically verifiable form of latent planning, (2) how this ability can be defined and measured, and (3) how this capability scales with model size.

The methodology first establishes a strict, two-condition causal definition of latent planning, distinguishing it from prior observational or probing-based work. For an LLM to be "latent planning," it must possess an internal representation of a future goal (a token or concept $t$) that:
1. Forward Planning: Causes the model to eventually generate $t$ (Condition 1).
2. Backward Planning: Causes the model to generate a preceding context that licenses $t$ (Condition 2).

To identify these causal mechanisms, the authors employ Transcoder Feature Circuits, a mechanistic interpretability technique. This method decomposes a model's dense MLP activations into sparse, monosemantic (interpretable) features and identifies the causal sub-graph (the "circuit") that explains a specific behavior.1 The study is conducted on the Qwen-3 family of open-source models, ranging from 0.6B to 14B parameters.

**Strengths:**

1. The paper's greatest strength is its insistence on a rigorous, two-condition causal definition of latent planning. This elevates the study from a correlational observation to a test of a mechanistic hypothesis. This strength is powerfully underscored by the refutation of probing-based methods in Appendix G, which demonstrates that high probing accuracy can be causally irrelevant.
2. The quality of the core experiment is extremely high. The a/an and is/are tasks serve as an elegant "minimal pair" testbed for planning. The causal interventions (ablation and boosting) in Section 4.4 provide "smoking gun" evidence for the discovered planning circuit. The analysis in Appendix E, which surgically separates task-solving ability from planning ability, is a brilliant and crucial piece of analysis that solidifies the paper's claims.

**Weaknesses:**

1. The complete failure of the methodology on the el/la task (Appendix D) is a significant weakness. The authors' explanation—that "Qwen-3 is not highly capable in language besides English and Chinese" —is an ad hoc hypothesis. This failure could alternatively imply that the "planning" mechanism found is not a general-purpose planning module at all, but a highly specific and brittle circuit for English grammatical agreement. This possibility severely undercuts the generality of the paper's claims.
2. The paper repeatedly claims that smaller models (4B-8B) have "nascent planning mechanisms"  but "fail" the task. It is unclear what this means mechanistically. Does the circuit exist, but is weak? Are some features missing? Does the model have the 'accountant' feature but lacks the causal connection to 'an' ? This "nascent" concept is central to the "emergence" narrative but remains poorly defined.

**Questions:**

On local planning, They say X features are described as "sensitive" and found in a "small minority." Is this evidence of a real, generalizable mechanism, or an artifact of steering on specific, polysemantic features that happen to fire on common n-grams? How could this mechanism be tested more robustly?

---

> ### Author Response · Authors · 2025-11-22
> **Author response [1/2]**
>
> Thank you for your thoughtful review! Here, we respond to your concerns:
> > The complete failure of the methodology on the el/la task (Appendix D) is a significant weakness. The authors' explanation—that "Qwen-3 is not highly capable in language besides English and Chinese" —is an ad hoc hypothesis. This failure could alternatively imply that the "planning" mechanism found is not a general-purpose planning module at all, but a highly specific and brittle circuit for English grammatical agreement. This possibility severely undercuts the generality of the paper's claims.
>
> We want to clarify that we don’t claim that planning is a general-purpose module: the successful backward planning we observe involves planning features connecting to downstream “say X” features, but our work shows that planning instances are learned separately. Whether models learn to perform a given instance of planning seems related to both model capacity and the frequency / complexity of the instance: thus, larger models learn to plan more, and models learn common phenomena (a/an) before complex and less common ones (rhyming couplets, or Spanish agreement).
>
> However, we agree that showing that successful planning is not limited to English grammatical agreement is interesting and valuable! So, we conducted a small follow-up experiment targeting a Chinese agreement phenomenon; few such phenomena exist, as Chinese has relatively sparse morphology. In particular, Chinese uses measure words, which function analogously to e.g. loaf (of bread) in English: rather than saying one bread, one says one loaf of bread. Similarly, in Chinese, one person becomes 一个人 (one [person-unit] person), while three pigs becomes 三头猪 (three [livestock-unit] pigs). Notably, different nouns require different measure words, but the measure word precedes the noun, much like a/an.
>
> So, we create graphs for 10 examples like 大森林里长着1000…棵树。 (In the large forest, there grew 1000…[measure-word] trees); 夜空中，他们看到了100…颗星星。 (In the night sky, they saw 100…[measure-word] stars); and 他看见四位调酒师端着四杯鸡尾酒 (He saw four bartenders carrying 4…[measure-word] cocktails). Note that these examples are machine translated, due to a lack of Chinese knowledge among the authors - this bottlenecked our ability to expand / scale up this experiment.
>
> We find, as in our English examples, that the model has planning features for “tree”, “star”, and “cocktail”. As in the previous experiments, we intervene by downweighting a given example’s planning feature, and also upweighting another’s. In doing so, we expect to elicit the other example’s measure word and planned word. These interventions are relatively effective; the interventions have effectiveness increasing with scale, causing the desired replacement 20% of the time for Qwen-3 (0.6B), and 50% of the time for Qwen-3 (14B).
>
> These interventions are especially effective in the case of an ambiguous planned token, e.g. the input 他看见四位农夫赶着四 (He saw four farmers with four…). There, the model activates both “cow” and “horse” features, but eventually outputs 头, the measure word for livestock. Upweighting “horse” features changes the measure word to agree with “horse”, instead.
>
> In these graphs, we find many “say [any measure word]” features, but fewer “say [specific measure word]” features. E.g., for the livestock measure word 头, “say head [of cattle]” features activate, but not all examples have these. Are there underlying “say [measure word] features that we miss due e.g. transcoder error? We test this via direct-effects experiments: if these fail, while full-effects experiments succeed, we know that intervening features exist in downstream MLPs, even if the transcoders miss them. We find that direct effects interventions work around 50% as well as our original replacement interventions, and that their effectiveness varies by measure word: in some examples, there is an intervening say [specific measure word]” feature, and in some there is not. We hypothesize that this is because some Chinese measure words have semantics that closely align with their corresponding noun.
>
> We have added details regarding this exploratory experiment to Appendix K.

---

> > ### Author Response · Authors · 2025-11-22
> > **Author response [2/2]**
> >
> > > The paper repeatedly claims that smaller models (4B-8B) have "nascent planning mechanisms" but "fail" the task. It is unclear what this means mechanistically. Does the circuit exist, but is weak? Are some features missing? Does the model have the 'accountant' feature but lacks the causal connection to 'an' ? This "nascent" concept is central to the "emergence" narrative but remains poorly defined.
> >
> > This is a good question, and one that we considered ourselves. There are a few points of potential breakage in the circuit:
> >
> > - The models could lack planning features entirely. This doesn’t appear to be the case in 4B/8B, because we can find and upweight these, as shown via our interventions
> > - The models could lack connections from planning features to a/an entirely. This also can’t be the case; if they lacked the connections entirely, then the either the all-effects interventions would fail (because there would be insufficient features to push the model towards the right answer) or the direct-effects interventions would perform equally to the all-effects one (because there are no a/an links anyway)
> >
> > So, the answer is that the circuit exists, but is weak. Still, it’s worth thinking about how a circuit could be weak?: do the models have too few planning features, or activate them too weakly?; or, do they have weak connections between these features and the “say a/an” features? Some combination of these is also possible.
> > We perform additional experiments on the a/an dataset, and find evidence that these models (4/8B) have relatively fewer planning features active on the an examples they get wrong. While they have an average of 18/23 features active on an examples they get correct, they have only 7 features active on incorrect an examples. By contrast, the smallest models have few planning features active overall, and the Qwen-3 (14B) has a much smaller gap in the number of features active in the two cases (21 vs. 16). So, while those features can be boosted
> > We have added this experiment to Appendix M.
> >
> > > On local planning, They say X features are described as "sensitive" and found in a "small minority." Is this evidence of a real, generalizable mechanism, or an artifact of steering on specific, polysemantic features that happen to fire on common n-grams? How could this mechanism be tested more robustly?
> >
> > We agree that the features on which we steer may correspond specifically to n-grams such as in/during the night. What we find compelling is the context in which these features appear: they fire when the model needs to predict an -ight rhyme, not just e.g. when the model needs to say a phrase starting with in. To us, this is the beginning of better planning: instead of just upweighting rhyming words, the model upweights fitting n-grams containing them, leading to more complex and coherent continuations. Moreover, there are clear ways for this mechanism to scale: as n grows, or as models move away from strict n-grams, their couplet continuations will grow more and more coherent. That is, real planning mechanisms and n-gram features may not be mutually exclusive.

---

> ### Author Response · Authors · 2025-11-28
>
> Hello Reviewer shSX - thanks again for your insightful review! If you have the time, we'd love to hear what you thought of our Chinese measure word experiment, and if that moves the needle on your thinking re: the generality and value of this planning work.

---

### Author Response · Authors · 2025-11-22
**Response to all reviewers**

Thank you all for your reviews - we are happy to have received such well-though-out comments. Here, we respond to questions shared by multiple reviewers; we have also added these points to a new Discussion section in the main text:

**Do our findings suggest a general planning mechanism?**: Our evidence suggests that the planning mechanisms we discover are not general in the sense that the model uses the exact same circuit for all planning tasks. Whether a given model plans on a given task is regulated by the model’s capacity, as well as the task’s complexity, along with its frequency and importance (in terms of training loss). Thus, larger models plan more, and common planning tasks are learned faster, resulting in piecemeal planning abilities, rather than a large set of abilities and a unified mechanism.

Successful planning circuits often follow a motif: there are planning features indicating the planned word, which then activate downstream features responsible for backward planning. However, models may learn to plan in one case (a/an agreement) but not others (couplets), simply because the former is more important to reducing its loss than the latter, and the model is too small to learn both.

 We believe that, even if they are not general, such mechanisms are still interesting, given planning's relevance to cognition and safety.

**These tasks are simple. Can this framework be extended to more complex tasks?**: These tasks are indeed simple, largely because the Qwen-3 models that we analyze struggle with complex planning tasks (like e.g. chain-of-thought unfaithfulness), preventing us from studying them. However, as open source models become more competent, our framework will still be able to explain these planning behaviors. Sometimes, as in CoT unfaithfulness examples where the model plans for a single-token answer, it may suffice to simply attribute from the given answer token, as done here.

In other cases, e.g. detecting a hidden goal that drives model behavior without producing one “smoking gun” answer token, we may want to attribute back from more general high-level model actions, such as “Why did the model produce a refusal?” or “Why did the model make a given suggestion?”. Though past work has not yet done so, attributing from such higher-level actions is possible by simply extending our framework, as one can attribute back from distributions over logits, or arbitrary directions in activation space. In this case, one must identify a direction in activation space corresponding to such an action, and attribute from this. We think that techniques like probing for such higher level actions, or using a difference-in-means-esque approach to identify causally relevant directions for such actions.

**These results come from only one model family. Will they generalize?**: We do study only one model family, as this allows us to analyze the effects of scale on planning. Moreover, no transcoders exist for similarly competent models (though more are in active development), and training a new transcoder suite for another, capable model family would be prohibitively expensive. That said, based on past work, we believe that the circuits we find may generalize: the circuits for e.g. multi-hop reasoning in Gemma-2 (2b) (Hanna et al., 2025) look similar to those in Claude (Lindsey et al., 2025), a vastly larger and more competent model.

On the methods side, as long as transcoders exist for a model, our methods should generalize to it. Even without transcoders, applying the forward/backward planning framework is in theory possible using probes on any model: one could train probes to extract a planned word or rhyme family. But, this would be much less flexible, requiring new probes for each task, and loses the fine-grained insights of transcoder feature circuits. Very recent work suggests that circuit-tracing may even be possible with neurons alone [1]. We thus believe that extending the current work to more models is possible, and would be excited to see follow-up work along these lines.

[1]: Arora et al. 2025, Language Model Circuits Are Sparse in the Neuron Basis

**Added Content**: We’ve added the following experiments to the appendix, as well as pointers to these appendices in the main text:
- An exploratory experiment showing that a planning phenomenon also occurs in the context of Chinese measure-word agreement. (Appendix K)
- Experiments clarifying what distinguishes nascent circuits from fully-formed ones. (Appendix M)
- Behavioral experiments showing that Qwen-3 matches the performance of instruction-tuned Qwen (the subject of our paper) on the a/an task, but underperforms it on the is/are task, indicating the importance of instruction-tuning. (Appendix J)
- Experiments showing that steering on planning features such as “accountant” does not produce the corresponding article “an” in all contexts; rather, steering generates “accountant” consistently, and “an” only when appropriate (Appendix L)

---

### Author Response · Authors · 2025-12-03
**Summary of Reviews and Responses**

As the response period comes to a close, we wanted to provide a summary of the reviews, as well as our responses to them. Overall, all reviewers found our paper sound (3/3/4) and our presentation good (3/3/4); most reviewers found the contributions excellent (4/2/4). Reviewers especially appreciated that "The quality of the core experiment is extremely high" (shSX), as well as our "clear and causally grounded definition of latent planning that distinguishes genuine planning from mere correlational predictability" (nfGc) and "interpretable scaling law that contributes to our understanding of model cognition" (JcTR).

However, as indicated by the overall scores (8/4/4), reviewers ShSX and nfGc in particular had concerns about some aspects of our paper. In response to these, we conducted a number of experiments and made meaningful edits to our paper. We outline both reviewer concerns and our responses here:

- **Generality of Planning Mechanism** (shSX, nfGc): We noted that the planning mechanisms we observe follow a general motif---planning features upweight words that license the planned token---but that there is no single planning mechanism across examples. We claim that this is an interesting contribution of our study: we find a jagged edge of planning abilities that emerge with scale. We added discussion of this to **Section 7**.
- **Generalization to More Complex Tasks** (nfGc, JcTR): We outlined how our techniques could be used to study complex, multi-step reasoning abilities in models---once we have sufficiently capable open weights models with transcoders trained on them. We added discussion of this to **Section 7**.
- **Cross-model generalization** (nfGc, JcTR): We noted that no other models both are capable of the tasks that we study and have the transcoders needed for our experiments; however, we highlighted how recent transcoder circuit results have transferred across model sizes and families. We added discussion of this to **Section 7**.
- **Lack of clarity about the definition of nascent planning mechanisms** (shSX): We ran an additional experiment showing that, in the case of models that possess planning features but fail, fewer planning features are present. Nascent planning circuits are thus ones in which models do not possess or insufficiently activate these planning features. We added this experiment to **Appendix M**.
- **Limited number and language of tasks** (shSX): We ran another experiment in Chinese, in which we tested for planning in the context of predicting measure words, and also found causally relevant features corresponding to planned words. We added this to **Appendix K**.
- **Unclear if effects come from correlated features** (nfGc): We added an experiment showing that steering on planning features in the a/an context does produce the planned word, but does not produce a/an in general; this is evidence against the correlated feature hypothesis. We added this to **Appendix L**.
- **Effects of instruction tuning** (nfGc): We ran a behavioral experiment showing that the base and IT'd models do have distinct planning abilities, however one is not consistently better than the other. We added this to **Appendix J**.
- **Usefulness of this paradigm for AI safety** (nfGc): We outlined how circuit-tracing could be used for monitoring models for hidden goals and other undesirable forms of planning in a real-life scenario.
- **Quantifying Context Licensing** (JcTR): We highlighted the ways in which we already quantify context licensing in **Section 5**, while also discussing the potential approaches for and challenges associated with additional quantification of the word-steering experiments.

We thank the reviewers for their contributions to the paper!

---

### Meta-Review · Area_Chair_edaL · 2025-12-09

**Summary:**

This paper discusses the ability of models to plan as they become larger, and also how such planning is represented in the internal representations. While the reviewer had some concerns, I believe they were addressed quite well by the authors, and explained well how their findings generalize, for example, to other tasks. Overall, I tend to believe the paper will be well-received by the community.

**Reviewer Concerns:**

Some of the general claims were not clear to reviewers.

**Reviewer Scores:**

I cannot predict that.

---

### Decision · Program_Chairs · 2026-01-26

Accept (Poster)